# MEDMKG: BENCHMARKING MEDICAL KNOWLEDGE EXPLOITATION WITH MULTIMODAL KNOWLEDGE GRAPH

## ABSTRACT

Medical deep learning models depend heavily on domain-specific knowledge to perform well on knowledge-intensive clinical tasks. Prior work has primarily leveraged unimodal knowledge graphs, such as the Unified Medical Language System (UMLS), to enhance model performance. However, integrating *multimodal* medical knowledge graphs remains largely underexplored, mainly due to the lack of resources linking imaging data with clinical concepts. To address this gap, we propose MEDMKG, a **Med**ical **M**ultimodal **K**nowledge **G**raph that unifies visual and textual medical information through a multi-stage construction pipeline. MEDMKG fuses the rich multimodal data from MIMIC-CXR with the structured clinical knowledge from UMLS, utilizing both rule-based tools and large language models for accurate concept extraction and relationship modeling. To ensure graph quality and compactness, we introduce Neighbor-aware Filtering (NaF), a novel filtering algorithm tailored for multimodal knowledge graphs. We evaluate MEDMKG across **five** tasks under **two** experimental settings, benchmarking **twenty-four** baseline methods and **four** state-of-the-art vision-language backbones on **six** datasets. Results show that MEDMKG not only improves performance in downstream medical tasks but also offers a strong foundation for developing adaptive and robust strategies for multimodal knowledge integration in medical artificial intelligence.

## 1 INTRODUCTION

Deep learning has demonstrated remarkable success in the medical domain, enabling tasks such as health risk prediction, disease diagnosis, and mortality forecasting (Esteva et al. (2019); Wang et al. (2024a); Miotto et al. (2018)). However, medical data often suffer from noise and missing values, limiting the effectiveness of feature representation learning. To address these challenges, researchers have increasingly integrated *unimodal* medical knowledge graphs into deep learning frameworks. These graphs offer structured and explicit representations of domain knowledge by encoding well-defined medical concepts and their relationships (Wang (2025); Qu (2022); Li et al. (2020); Wu et al. (2023)). Incorporating such structured knowledge has led to notable improvements in different tasks, including health risk prediction (Ye et al. (2021); Choi et al. (2016); Ma et al. (2020)), adverse drug reaction prediction (Wang et al. (2021); Zhang et al. (2021); Bean et al. (2017)), and medical coding (Luo et al. (2024); Shi et al. (2017)).

Nevertheless, many important clinical tasks require **multimodal** data as model inputs, such as medical visual question answering (VQA) (Lin et al. (2023)) and text-image retrieval (Kitanovski et al. (2017)). Relying solely on unimodal medical knowledge graphs in these contexts often fails to yield significant performance gains, due to the absence of explicit relationships between visual data and medical concepts. This limitation has hindered the ability of current multimodal deep learning models to fully capitalize on domain knowledge in knowledge-intensive tasks. Addressing this gap necessitates the development of a comprehensive multimodal medical knowledge graph. However, building such a resource introduces the following critical challenges:

- **C1: Quality Concern**. A multimodal medical knowledge graph must be of high quality and practical utility. This includes the accurate identification and representation of diverse intra- and

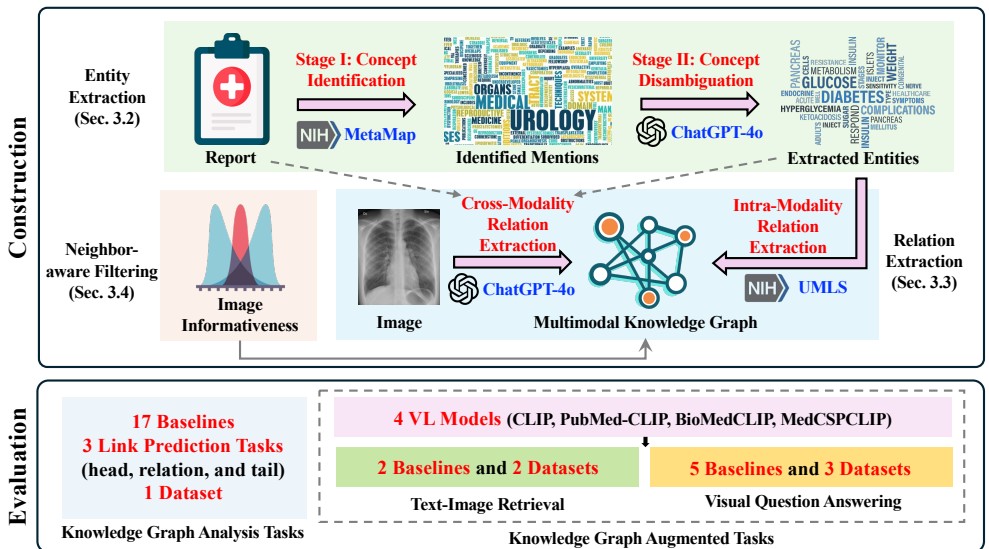

Figure 1: The overview of MEDMKG construction pipeline and evaluation methods.

inter-modal relationships, which requires a carefully designed and systematically implemented construction process.

- **C2: Utility Concern**. Beyond quality, it is essential to evaluate whether the graph can effectively enhance model performance on downstream tasks. The graph must encode clinically meaningful multimodal knowledge that directly supports a wide range of knowledge-intensive applications.

To bridge this research gap and address the identified challenges, we introduce MEDMKG, a **Med**ical **M**ultimodal **K**nowledge **G**raph that unifies visual and textual medical information. To tackle **C1** (*Quality Concern*), as shown in Figure 1, we develop a multi-stage construction pipeline that ensures high-fidelity cross-modal integration by combining the rich visual and textual information in MIMIC-CXR (Johnson et al. (2019)) with the structured clinical knowledge in the Unified Medical Language System (UMLS) (Bodenreider (2004)). Our method leverages the domain accuracy of rule-based tools together with the contextual reasoning capabilities of large language models (LLMs), enabling precise extraction of clinical concepts and their relationships. To further ensure conciseness and informativeness, we propose a simple yet effective Neighbor-aware Filtering algorithm (NaF) to enhance the quality of MEDMKG by ranking and filtering medical images. Both expert qualitative evaluations and quantitative benchmarking validate that MEDMKG achieves high quality and is well-suited for practical downstream use.

To address **C2** (*Utility Concern*), we conduct extensive experiments across two complementary settings to demonstrate the practical utility of MEDMKG, as shown Figure 1. First, in the setting of knowledge graph analysis, we assess the intrinsic quality of MEDMKG through a link prediction task. Second, in the setting of knowledge graph augmentation, we integrate MEDMKG into downstream applications, including medical text-image retrieval and visual question answering. Our comprehensive evaluation spans 24 baselines, 4 vision-language backbones, and 6 datasets covering 5 distinct tasks. This broad evaluation framework allows us to systematically explore how MEDMKG contributes to downstream performance. From these experiments, we derive several key insights:

- *Model Choice Should Align with Graph Structure*: Effective modeling of multimodal medical knowledge graphs requires selecting well-suited network architectures to handle their heterogeneous and relational nature, underscoring the importance of matching model design to graph characteristics.

- *External Knowledge Improves Downstream Tasks*: Incorporating structured medical knowledge consistently enhances downstream applications such as image–text retrieval and visual question answering, though the extent of improvement depends on the integration strategy and the underlying model architecture.

- *Balancing Knowledge Integration and Model Robustness*: While external knowledge generally improves coverage and reasoning capability, it also introduces challenges related to precision, recall and overfitting, highlighting the need for selective and adaptive knowledge fusion mechanisms.

- *Future Work Needs Unified and Adaptive Frameworks*: Advancing the field will require developing integration strategies that are both backbone-agnostic and adaptable, enabling knowledge graphs to be leveraged effectively across pretraining and fine-tuning stages for robust, generalizable improvements.

In summary, our contributions are threefold:

- **Construction of MEDMKG**: We present MEDMKG, a new medical multimodal knowledge graph that integrates clinical terminology and visual instances, providing a crucial resource for the development of knowledge-intensive multimodal models.

- **Effective Multimodal Knowledge Graph Filtering Algorithm**: We introduce Neighbor-aware Filtering (NaF), a targeted metric for ranking and filtering images in the context of a multimodal knowledge graph, which helps maintain the graph's quality and conciseness.

- **Extensive Benchmarking**: We conduct comprehensive evaluations spanning 5 tasks, 2 experimental settings, 24 baseline methods, 4 vision-language backbones, and 6 diverse datasets. Our results demonstrate that MEDMKG meaningfully improves performance on knowledge-intensive medical applications and opens the door to new adaptive fusion strategies in multimodal learning.

## 2 RELATED WORK

**Multimodal Learning in the Medical Domain.** Multimodal learning has seen widespread application in various medical tasks, including criticality prediction (Wang et al. (2023a; 2024c); Xu et al. (2018); Zhong et al. (2024); Feng et al. (2019); Tang et al. (2020)), readmission prediction (Yang & Wu (2021); Wang et al. (2023a; 2024c)), adverse drug reaction prediction (Luo et al. (2023)), and medical visual question answering (Li et al. (2024); Moor et al. (2023); Wang et al. (2024b;d)). Despite their success, most current multimodal methods in the medical domain are predominantly data-driven and rely on task-specific datasets rather than leveraging explicit, structured knowledge. This reliance limits their effectiveness in addressing knowledge-intensive tasks and highlights the need for developing robust, knowledge-reliable approaches and benchmarks.

**Medical Knowledge Graphs.** Medical knowledge graphs have become indispensable for organizing and interpreting complex biomedical data. Traditional medical knowledge bases have provided critical insights across both comprehensive systems (Donnelly et al. (2006); Bodenreider (2004); Lipscomb (2000)) and specialized domains (Wishart et al. (2006); Goh et al. (2007)). These systems are typically built through extensive manual annotation, long development cycles, and the sustained involvement of domain experts. However, the labor-intensive nature of annotating medical imaging data presents significant challenges when attempting to generalize these approaches to the construction of multimodal knowledge graphs. To address scalability concerns, several automated methods have been proposed for building medical knowledge graphs. Some works focus on constructing comprehensive graphs (Lin et al. (2015); Chandak et al. (2023)), while others target specific subdomains, such as pharmacology (Bean et al. (2017); Zhang et al. (2021); Wang et al. (2021)), broader biomedical fields (Vlietstra et al. (2017); Fei et al. (2021); Yuan et al. (2020)), Covid-19 (Michel et al. (2020)), etc. Although these automated approaches offer improved efficiency, they often rely on overly simplified or outdated techniques that compromise accuracy.

**Multimodal Knowledge Graphs.** Recent research has begun to extend traditional unimodal knowledge graphs into the multimodal realm. Existing approaches for constructing multimodal knowledge graphs typically utilize search engines (Wang et al. (2020); Zhang et al. (2022); Liu et al. (2019)), web crawlers (Wang et al. (2023c); Oñoro-Rubio et al. (2017)), or queries to open-source knowledge bases such as Wikipedia (Wang et al. (2020); Zhang et al. (2022)). While these methods perform adequately in general domains where cross-modal alignment is often achievable, the inherent limitations in retrieval accuracy can adversely affect the quality of medical knowledge graphs. This challenge is particularly pronounced in the medical domain, where precision and reliability are paramount.

## 3 CONSTRUCTION OF MEDMKG

### 3.1 PROBLEM FORMULATION

Constructing a multimodal radiological knowledge graph from scratch poses significant challenges due to the scale, complexity, and heterogeneity of data modalities. A more practical and reliable strategy is to extend an existing unimodal knowledge graph by systematically incorporating additional modalities. In this work, we formulate the construction of our multimodal radiological knowledge graph as a *modality-wise graph extension* problem.

We begin with the Unified Medical Language System (UMLS) (Bodenreider (2004)), a comprehensive biomedical knowledge base that standardizes and interconnects diverse health-related vocabularies via concept unique identifiers (CUIs). UMLS offers a rich repository of medical concepts and semantic relationships, serving as the foundational backbone for structured medical knowledge integration. For example, the clinical relation "*aspirin is used to treat myocardial infarction*" is represented as a triplet (C0011849, treats, C0020538), where "C0011849" corresponds to "*Aspirin*" and "C0020538" to "*Myocardial Infarction (Heart Attack)*".

We expand the UMLS graph by introducing radiological image nodes and establishing cross-modal edges. The resulting graph contains two types of nodes: (1) **clinical concepts**, inherited directly from UMLS, and (2) **radiological images**. It also includes two types of edges: (1) **intra-modality edges** among clinical concepts (as defined in UMLS), and (2) **cross-modality edges** that link clinical concepts to corresponding images.

To perform the multimodal extension, we leverage the MIMIC-CXR dataset (Johnson et al. (2019)), which consists of paired radiology reports and chest X-ray images. Details about the preprocessing of MIMIC-CXR is available in Appendix E.1. From each report, we extract relevant clinical concepts and align them with their associated images, thereby establishing meaningful cross-modal connections. This design enables the extended knowledge graph to seamlessly integrate textual and visual medical information within a unified and structured framework.

### 3.2 CONCEPT EXTRACTION

A central challenge in constructing MEDMKG lies in accurately establishing cross-modal edges between radiological images and clinical concepts. To address this, we design a two-stage pipeline that leverages the complementary strengths of rule-based systems and large language models (LLMs). Rule-based tools are highly effective in handling clinical terminologies and ontologies, offering broad coverage of domain-specific entities. In contrast, LLMs provide strong contextual understanding and disambiguation capabilities, enabling more accurate interpretation of report-level semantics. By integrating these two approaches, our pipeline achieves both the comprehensive coverage and semantic precision necessary for high-quality concept extraction and reliable cross-modal alignment.

**Stage I – Concept Identification.** We begin by applying MetaMap (Aronson & Lang (2010)), a widely used rule-based tool, to each radiology report to identify candidate mentions of UMLS concepts. This step produces an exhaustive set of potential concept mappings for each mention, ensuring comprehensive coverage of clinically relevant entities. To focus on concepts with clinical significance, we filter out irrelevant semantic types based on domain knowledge. A complete list of excluded semantic types is provided in Appendix E.2.

**Stage II – Concept Disambiguation.** Next, we refine the candidate concepts using ChatGPT-4o (OpenAI Achiam et al. (2023)) that considers both the full radiology report and the list of extracted candidates. For each mention, the LLM is prompted to select the most contextually appropriate concept, leveraging its strong semantic understanding to resolve ambiguity. This stage eliminates spurious or out-of-context candidates, resulting in a clean and accurate set of disambiguated clinical concepts aligned with each image.

This two-stage design enables precise and context-aware mapping of clinical concepts to radiological images, ensuring the construction of high-quality cross-modal edges in the resulting knowledge graph. Aggregating the selected concepts across all mentions in a report yields the final set of clinical concepts associated with each image.

## 3.3 Relation Extraction

With the clinical concepts identified, we further enrich the knowledge graph by establishing relations:

**Intra-Modality Relations.** We introduce edges between identified clinical concepts whenever a relation is defined between them in UMLS. Only validated relations connecting distinct concepts are added, ensuring that intra-modality relationships are medically accurate and standardized.

**Cross-Modality Relations.** Each image is linked to its extracted clinical concepts through cross-modality edges. However, beyond simply linking images and concepts, we also assign a semantic label to each edge to reflect the nature of the relationship. Specifically, each relation is categorized as *Positive*, *Negative*, or *Uncertain*, indicating whether the concept is supported by, contradicted by, or ambiguously discussed in the corresponding report.

While the intra-modality relations are extracted through querying the UMLS knowledge base, the cross-modal relation extraction is performed jointly with concept disambiguation. During the LLM prompting process, the model is additionally instructed to assess the semantic stance (positive, negative, or uncertain) between the image and each concept. These relation labels are used to annotate the edges accordingly. Details concerning the prompting procedure are available in Appendix E.3, while the analysis on the selection of LLM can be found in Appendix E.4.

## 3.4 Neighbor-Aware Filtering for Image Informativeness

The full construction process produces a highly comprehensive multimodal knowledge graph. However, its large scale, with numerous images and associated concepts, creates challenges for storage, computation, and downstream analysis. In particular, many radiological images are *redundant* because they capture similar and homogeneous regions (Zhou et al. (2010)). This redundancy can overwhelm subsequent analysis and reduce graph efficiency. To improve efficiency without sacrificing knowledge quality, we introduce a filtering strategy that prioritizes the most informative and distinctive images.

Ideally, a representative medical image should be connected to multiple clinical concepts through diverse relations, making the number of its neighboring nodes a key indicator of informativeness. However, relying solely on the number of neighbors may introduce noise, as some medical concepts are linked to a large number of generic or non-discriminative images. To mitigate this, we additionally consider the distinctiveness of an image in the context of its 2-hop neighborhood. Intuitively, if a relation–concept pair is associated with only a few images, those images are likely to carry more unique and clinically informative content.

Based on this insight, we propose a **Neighbor-aware Filtering (NaF)** strategy that balances both connectivity and distinctiveness. The informativeness score of an image $m$ is defined as:

$$\text{NaF}(m) = \sum_{(r,c) \in \mathcal{N}_m} \log \frac{M}{|\mathcal{N}_{(r,c)}|}, \tag{1}$$

where each triplet $(m, r, c)$ represents a connection between image $m$, relation $r$, and concept $c$; $\mathcal{N}_m$ denotes the 1-hop neighbors of $m$; $M$ represents the number of medical images in the knowledge graph; and $\mathcal{N}_{(r,c)}$ is the set of images linked to concept $c$ via relation $r$.

By combining these two dimensions, the designed NaF strategy effectively prioritizes images that are both rich in clinical content and contribute unique, informative knowledge to the graph. After computing the informativeness scores, we rank all images in descending order and select them from top to bottom until the full set of concepts is covered. This strategy ensures that the final graph retains maximal clinical richness and diversity while eliminating redundant or overly generic images, thereby improving scalability and downstream utility. More details of the NaF strategy algorithm are available in Appendix E.5.

## 3.5 Quantitative and Qualitative Analysis

To acquire an intuitive understanding of MEDMKG's statistical characteristics and soundness of MEDMKG, we performed both quantitative and qualitative analyses.

**Quantitative Analysis.** MEDMKG's statistics are detailed in Table 1. The moderate scale of MEDMKG facilitates convenient utilization in diverse application scenarios with different com-

Table 1: Data Statistics Summary

| Statistic | Count |
|---|---|
| Total Number of Edges | 35,387 |
| Number of Concepts | 3,149 |
| Number of Images | 4,868 |
| Number of Relations | 262 |
| Number of Cross-modality Edges | 20,705 |
| Number of Intra-modality Edges | 14,682 |
| Image-to-Concept Ratio | 1.55 |
| Average Edges per Image | 4.25 |
| Average Edges per Concept | 11.24 |

Figure 2: Human assessment results.

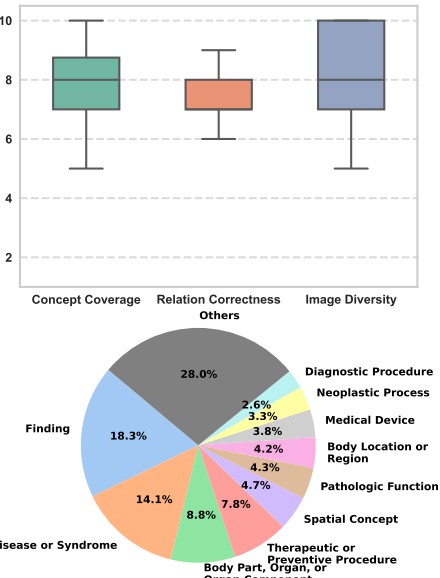

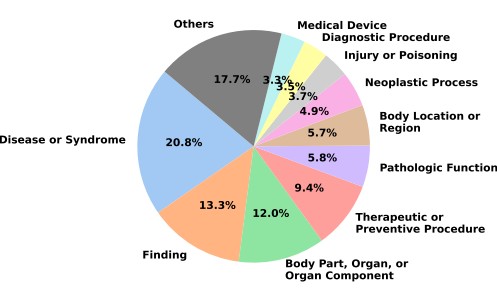

(a) Distribution of Head Concepts per Semantic Types

(b) Distribution of Tail Concepts per Semantic Types

Figure 3: Distribution of entities involved in MEDMKG. The top 10 semantic types are shown individually, and rare types are grouped as "Others."

putational budgets. Additionally, images and concepts are intensively connected with intra- and cross-modal neighbors, promoting rich multimodal reasoning. Furthermore, Figure 3 shows the distribution of semantic types between the clinical concepts involved, indicating a broad and balanced coverage of the areas of clinical knowledge.

**Qualitative Analysis.** To further assess the quality of MEDMKG, we conducted a human evaluation with experienced radiologists. The experts reviewed a set of sampled subgraphs and assigned quality scores across three key dimensions, each rated on a scale from 1 to 10: (1) *concept coverage* — whether the graph captures the key image-related clinical concepts; (2) *relation correctness* — whether the cross-modal relations are accurately identified; and (3) *image diversity* — whether the linked images reflect a broad range of clinical scenarios. Higher scores indicate better performance on each metric. As illustrated in Figure 2, MEDMKG achieves an average of approximately 80% across all three metrics. Compared with previous studies where 60% of agreement is regarded convincing (Schäfer et al. (2024); Kilicoglu et al. (2008)), this result indicates MEDMKG's reliability and practical utility as a multimodal medical knowledge source. Further details on the evaluation protocol are provided in Appendix F. An illustration of the constructed MEDMKG is available in Appendix E.6.

## 4 BENCHMARK

We evaluate MEDMKG under two complementary scenarios: **knowledge graph analysis** and **knowledge graph augmentation**. In the knowledge graph analysis setting, we assess tasks that directly utilize the internal structure and semantics of the graph, i.e., *link prediction*. In the knowledge graph augmentation setting, MEDMKG is employed as auxiliary knowledge to enhance the performance of external multimodal applications, including *multimodal text-image retrieval* and *multimodal visual question answering (VQA)*.

### 4.1 LINK PREDICTION

The link prediction tasks (Bordes et al. (2013)) focus on inferring missing links between entities by predicting either the head entity, the tail entity, or the relation connecting them. Specifically, given two known components of a triple, such as a relation and one entity, or two entities, the goal is to

Table 2: Performance of 17 approaches on three link prediction tasks (mean$_{\pm\text{std}}$).

| Model | Head Prediction | | | | Relation Prediction | | | | Tail Prediction | | | |
|---|---|---|---|---|---|---|---|---|---|---|---|---|
| | MR $\downarrow$ | Hits@3 $\uparrow$ | Hits@5 $\uparrow$ | Hits@10 $\uparrow$ | MR $\downarrow$ | Hits@3 $\uparrow$ | Hits@5 $\uparrow$ | Hits@10 $\uparrow$ | MR $\downarrow$ | Hits@3 $\uparrow$ | Hits@5 $\uparrow$ | Hits@10 $\uparrow$ |
| TransR | $1505.66_{\pm36.95}$ | $1.80_{\pm0.33}$ | $3.71_{\pm0.49}$ | $7.50_{\pm0.40}$ | $106.01_{\pm5.05}$ | $5.84_{\pm0.59}$ | $10.55_{\pm0.62}$ | $19.98_{\pm1.27}$ | $887.07_{\pm33.44}$ | $3.34_{\pm0.27}$ | $6.65_{\pm0.34}$ | $13.11_{\pm0.27}$ |
| TransD | $1219.92_{\pm34.04}$ | $3.84_{\pm0.60}$ | $7.56_{\pm0.87}$ | $12.22_{\pm1.18}$ | $53.49_{\pm9.96}$ | $27.49_{\pm4.21}$ | $35.52_{\pm4.49}$ | $46.36_{\pm4.66}$ | $586.61_{\pm15.25}$ | $5.25_{\pm0.48}$ | $10.90_{\pm0.53}$ | $18.67_{\pm0.67}$ |
| TransE | $1248.36_{\pm70.15}$ | $3.36_{\pm0.64}$ | $6.33_{\pm1.20}$ | $9.93_{\pm1.92}$ | $39.74_{\pm1.48}$ | $20.13_{\pm1.16}$ | $28.79_{\pm0.99}$ | $42.45_{\pm0.88}$ | $544.79_{\pm32.35}$ | $4.99_{\pm0.80}$ | $9.20_{\pm1.59}$ | $15.14_{\pm2.66}$ |
| TransH | $1263.25_{\pm75.38}$ | $3.29_{\pm0.88}$ | $6.16_{\pm1.38}$ | $9.99_{\pm1.68}$ | $37.40_{\pm0.37}$ | $22.02_{\pm1.73}$ | $30.21_{\pm1.10}$ | $43.41_{\pm1.02}$ | $561.39_{\pm31.90}$ | $5.05_{\pm0.50}$ | $9.67_{\pm1.25}$ | $16.07_{\pm1.92}$ |
| RotatE | $1560.49_{\pm55.42}$ | $1.35_{\pm0.67}$ | $2.69_{\pm0.91}$ | $5.04_{\pm1.41}$ | $129.84_{\pm2.36}$ | $0.77_{\pm0.15}$ | $1.42_{\pm0.26}$ | $3.25_{\pm0.41}$ | $739.95_{\pm28.29}$ | $1.54_{\pm0.61}$ | $3.47_{\pm1.09}$ | $6.82_{\pm2.06}$ |
| DistMult | $3590.63_{\pm376.55}$ | $0.10_{\pm0.14}$ | $0.19_{\pm0.27}$ | $0.38_{\pm0.44}$ | $119.13_{\pm12.36}$ | $2.17_{\pm2.61}$ | $3.92_{\pm4.33}$ | $6.95_{\pm6.18}$ | $3582.56_{\pm386.85}$ | $0.15_{\pm0.17}$ | $0.26_{\pm0.31}$ | $0.50_{\pm0.64}$ |
| SimplE | $4032.57_{\pm44.70}$ | $0.01_{\pm0.01}$ | $0.04_{\pm0.03}$ | $0.10_{\pm0.04}$ | $133.16_{\pm1.43}$ | $0.63_{\pm0.14}$ | $1.19_{\pm0.18}$ | $2.95_{\pm0.35}$ | $4033.27_{\pm42.33}$ | $0.05_{\pm0.03}$ | $0.06_{\pm0.04}$ | $0.11_{\pm0.03}$ |
| TuckER | $1533.74_{\pm80.37}$ | $2.83_{\pm0.48}$ | $4.51_{\pm0.93}$ | $7.45_{\pm1.61}$ | $43.75_{\pm4.73}$ | $46.67_{\pm1.44}$ | $55.42_{\pm2.18}$ | $64.01_{\pm2.32}$ | $1235.88_{\pm134.15}$ | $4.31_{\pm0.25}$ | $7.12_{\pm0.53}$ | $11.71_{\pm1.21}$ |
| ComplEx | $3790.54_{\pm401.89}$ | $0.09_{\pm0.14}$ | $0.20_{\pm0.35}$ | $0.31_{\pm0.47}$ | $125.58_{\pm12.61}$ | $0.93_{\pm1.58}$ | $1.81_{\pm2.74}$ | $3.79_{\pm4.72}$ | $3782.86_{\pm406.27}$ | $0.12_{\pm0.16}$ | $0.23_{\pm0.38}$ | $0.40_{\pm0.63}$ |
| RESCAL | $3849.47_{\pm125.08}$ | $0.03_{\pm0.04}$ | $0.06_{\pm0.05}$ | $0.12_{\pm0.09}$ | $127.51_{\pm3.82}$ | $0.44_{\pm0.11}$ | $0.99_{\pm0.38}$ | $2.40_{\pm0.54}$ | $3845.42_{\pm123.70}$ | $0.02_{\pm0.03}$ | $0.03_{\pm0.04}$ | $0.07_{\pm0.06}$ |
| HypER | $3564.84_{\pm584.55}$ | $0.33_{\pm0.39}$ | $0.56_{\pm0.67}$ | $0.93_{\pm1.10}$ | $122.79_{\pm13.72}$ | $1.16_{\pm0.77}$ | $2.20_{\pm1.48}$ | $4.12_{\pm2.48}$ | $2933.17_{\pm1268.15}$ | $1.07_{\pm1.39}$ | $1.71_{\pm2.16}$ | $2.89_{\pm3.54}$ |
| ConvE | $2071.83_{\pm130.51}$ | $1.81_{\pm0.09}$ | $2.76_{\pm0.23}$ | $4.59_{\pm0.46}$ | $59.35_{\pm1.42}$ | $18.79_{\pm2.08}$ | $26.51_{\pm1.77}$ | $36.10_{\pm1.45}$ | $777.91_{\pm29.16}$ | $4.28_{\pm0.50}$ | $6.87_{\pm0.62}$ | $11.03_{\pm1.05}$ |
| ConvR | $2438.14_{\pm105.67}$ | $0.62_{\pm0.12}$ | $1.05_{\pm0.23}$ | $1.77_{\pm0.29}$ | $78.55_{\pm2.80}$ | $6.12_{\pm1.25}$ | $10.17_{\pm1.72}$ | $17.75_{\pm2.09}$ | $787.58_{\pm50.12}$ | $2.25_{\pm0.25}$ | $3.80_{\pm0.29}$ | $6.83_{\pm0.46}$ |
| AttH | $2113.85_{\pm717.20}$ | $0.36_{\pm0.30}$ | $0.86_{\pm0.75}$ | $1.66_{\pm1.46}$ | $31.69_{\pm12.50}$ | $26.94_{\pm11.39}$ | $36.91_{\pm11.65}$ | $50.45_{\pm11.41}$ | $523.86_{\pm7.46}$ | $5.72_{\pm0.89}$ | $8.90_{\pm1.19}$ | $14.10_{\pm1.58}$ |
| MurE | $1248.36_{\pm70.15}$ | $3.36_{\pm0.64}$ | $6.33_{\pm1.20}$ | $9.93_{\pm1.92}$ | $39.74_{\pm1.48}$ | $20.13_{\pm1.16}$ | $28.79_{\pm0.99}$ | $42.45_{\pm0.88}$ | $544.79_{\pm32.35}$ | $4.99_{\pm0.80}$ | $9.20_{\pm1.59}$ | $15.14_{\pm2.66}$ |
| MurP | $2771.43_{\pm1234.52}$ | $2.14_{\pm2.11}$ | $4.29_{\pm4.29}$ | $7.00_{\pm6.84}$ | $158.56_{\pm57.45}$ | $3.45_{\pm3.86}$ | $4.67_{\pm4.65}$ | $7.02_{\pm6.06}$ | $590.11_{\pm49.63}$ | $4.34_{\pm0.82}$ | $7.59_{\pm1.61}$ | $12.64_{\pm3.27}$ |
| NTN | $4007.11_{\pm63.65}$ | $0.01_{\pm0.01}$ | $0.02_{\pm0.03}$ | $0.06_{\pm0.04}$ | $140.38_{\pm9.31}$ | $0.14_{\pm0.11}$ | $0.29_{\pm0.25}$ | $1.02_{\pm0.92}$ | $3994.01_{\pm76.63}$ | $0.01_{\pm0.02}$ | $0.02_{\pm0.02}$ | $0.05_{\pm0.05}$ |

predict the missing element that completes the triple. This task helps improve the completeness and utility of knowledge graphs by filling in missing entities or relations between entities.

**Baselines, Evaluation Metrics & Implementation.** We benchmark 17 widely-used link prediction models on our constructed KG, grouped into the following representative categories: (1) *Translation-based models*: TransE (Bordes et al. (2013)), TransH (Wang et al. (2014)), TransR (Lin et al. (2015)), TransD (Ji et al. (2015)) and RotatE (Sun et al. (2019)). (2) *Tensor factorization models*: RESCAL (Nickel et al. (2011)), DistMult (Yang et al. (2014)), ComplEx (Trouillon et al. (2016)), SimplE (Kazemi & Poole (2018)), and TuckER (Balažević et al. (2019b)). (3) *Convolution-based models*: HypER (Balažević et al. (2019a)), ConvE (Dettmers et al. (2018)), and ConvR (Jiang et al. (2019)). (4) *Manifold-based models*: AttH (Chami et al. (2020)), MurP (Balazevic et al. (2019)), and MurE (Balazevic et al. (2019)). (5) *Neural tensor model*: NTN (Socher et al. (2013)). More details about these baselines can be found in Appendix G.1. We evaluate the performance of the models using widely accepted metrics for link prediction, namely Mean Rank (MR), and Hits@$K$ (with $K$ set to 3, 5, and 10). Detailed descriptions of these metrics are provided in Appendix G.2. All models are optimized using the AdamW optimizer (Loshchilov et al. (2017)) with a batch size of 2,048 and a learning rate of 0.001. The training is run for a maximum of 500 epochs with an early stopping mechanism (patience set to 5 epochs) to prevent overfitting. Data are split into training, validation, and test sets with an 8:1:1 ratio.

**Evaluation Results.** Table 2 reports the performance of 17 link prediction baselines across head, relation, and tail tasks on our MEDMKG. A clear performance gap emerges between head and tail prediction: models achieve higher Hits@$K$ scores and lower mean ranks on tail entities, which are exclusively clinical concepts, while head entities combine images and concepts. This heterogeneity makes head prediction more challenging, as models must align multimodal representations within a shared embedding space. Among the baselines, translation-based models (TransD, TransE, TransH) achieve the strongest overall results, with TransD yielding the best Hits@10 across head, relation, and tail prediction. In contrast, tensor factorization models show mixed outcomes: while TuckER performs relatively well on relation prediction, others (e.g., SimpIE, RESCAL) perform poorly, indicating limited and inconsistent effectiveness in entity linking. These findings emphasize the importance of selecting models that align with the multimodal and relational structure of medical knowledge graphs. To enhance the overall capability of knowledge graph representation learning, future work may explore combining translation-based and tensor factorization-based models to leverage their complementary strengths, and proposing modality-aware link prediction module to aid the performance in head prediction task.

## 4.2 KNOWLEDGE-AUGMENTED TEXT-IMAGE RETRIEVAL

The knowledge-augmented text-image retrieval task aims to enhance conventional medical text-image retrieval (Demner-Fushman et al. (2012)) by leveraging domain knowledge encoded in a multimodal medical knowledge graph.

**Datasets & Backbone Models.** We leverage two representative datasets for the medical text-image retrieval task, i.e., OpenI (Demner-Fushman et al. (2016)) and MIMIC-CXR (Johnson et al. (2019)), following previous work (Wang et al. (2024c)). To prevent any potential data leakage regarding MIMIC-CXR, we only select text-image pairs that were not used during the curation of MEDMKG,

Table 3: Results (%) on Text-image Retrieval Task for OpenI and MIMIC-CXR Datasets. Metrics highlighted with green indicate improvement over backbone, while red refers to drop. Notable augmentation with MEDMKG is observed, especially for KnowledgeCLIP.

| Methods | OpenI | | | | | | MIMIC-CXR | | | | | |
|---|---|---|---|---|---|---|---|---|---|---|---|---|
| | Precision @$K$ ↑ | | | Recall @$K$ ↑ | | | Precision @$K$ ↑ | | | Recall @$K$ ↑ | | |
| | 10 | 20 | 100 | 10 | 20 | 100 | 10 | 20 | 100 | 10 | 20 | 100 |
| **CLIP** | 1.17 | 1.00 | 0.56 | 11.10 | 19.24 | 53.48 | 1.11 | 0.98 | 0.58 | 11.11 | 19.52 | 58.26 |
| + FashionKLIP | 1.29 | 1.16 | 0.63 | 12.64 | 22.75 | 60.46 | 1.19 | 0.99 | 0.56 | 11.91 | 19.82 | 56.06 |
| + KnowledgeCLIP | 2.63 | 1.99 | 0.79 | 25.56 | 38.83 | 76.16 | 2.33 | 1.73 | 0.74 | 23.32 | 34.53 | 74.37 |
| **PubMedCLIP** | 1.17 | 0.98 | 0.51 | 10.81 | 18.47 | 48.46 | 0.69 | 0.65 | 0.43 | 6.91 | 13.01 | 42.79 |
| + FashionKLIP | 1.54 | 1.21 | 0.70 | 15.10 | 23.38 | 67.73 | 0.73 | 0.72 | 0.49 | 7.31 | 14.41 | 49.20 |
| + KnowledgeCLIP | 1.49 | 1.17 | 0.61 | 14.33 | 22.61 | 59.41 | 1.26 | 1.13 | 0.60 | 12.61 | 22.62 | 59.96 |
| **BiomedCLIP** | 1.04 | 0.79 | 0.42 | 9.90 | 15.10 | 40.45 | 2.02 | 1.59 | 0.66 | 20.12 | 31.63 | 65.77 |
| + FashionKLIP | 1.46 | 1.15 | 0.60 | 14.33 | 22.47 | 58.22 | 2.02 | 1.49 | 0.68 | 20.12 | 29.63 | 67.77 |
| + KnowledgeCLIP | 1.26 | 0.95 | 0.49 | 12.50 | 18.61 | 47.54 | 2.64 | 1.94 | 0.71 | 26.33 | 38.74 | 70.77 |
| **MedCSPCLIP** | 1.60 | 1.10 | 0.54 | 15.73 | 21.35 | 52.14 | 3.77 | 2.59 | 0.82 | 37.69 | 51.65 | 81.58 |
| + FashionKLIP | 1.81 | 1.36 | 0.60 | 17.84 | 26.54 | 57.65 | 4.02 | 2.69 | 0.85 | 40.19 | 53.75 | 84.98 |
| + KnowledgeCLIP | 1.90 | 1.40 | 0.62 | 18.61 | 27.18 | 59.55 | 4.95 | 3.14 | 0.89 | 49.50 | 62.66 | 88.99 |

and we randomly sample a fixed set of 10,000 pairs from these remaining examples. Since no predefined splits exist, both datasets are divided into training, validation, and test sets with an 8:1:1 ratio. To comprehensively assess the impact of knowledge augmentation, we employ four open-sourced vision–language models as backbones: CLIP (Radford et al. (2021)), PubMedCLIP (Eslami et al. (2023)), BioMedCLIP (Zhang et al. (2023)), and MedCSPCLIP (Wang et al. (2024c)). Additional details about these models are available in Appendix H.

**Baselines, Evaluation Metrics & Implementation.** For benchmarking, we consider two knowledge-augmented retrieval methods: KnowledgeCLIP (Pan et al. (2022)) and FashionKLIP (Wang et al. (2023b)). More information about these baselines is available in Appendix I.1. We comprehensively evaluate retrieval performance using standard metrics, i.e., precision@$K$ and recall@$K$, with $K$ set to 10, 20, and 100. Detailed metric descriptions can be found in Appendix I.2. All models are optimized using the AdamW optimizer (Loshchilov & Hutter (2017)). The hidden state dimension is uniformly set to 512, and the learning rate is configured to 0.0001. Training is conducted for a maximum of 30 epochs with an early-stopping patience of 3 epochs.

**Evaluation Results.** Table 3 shows that knowledge augmentation consistently improves retrieval performance across both OpenI and MIMIC-CXR, particularly in low-K settings. This indicates that external knowledge enhances the model's ability to identify the most relevant matches at top ranks. Among the two strategies, KnowledgeCLIP (postraining-based) shows strong and consistent gains across most settings, especially on MIMIC-CXR, while FashionKLIP (joint fine-tuning) provides more noticeable improvements on OpenI relative to its effect on MIMIC-CXR. The overall trend suggests that integrating external knowledge, whether through pretraining or joint fine-tuning, can significantly benefit medical retrieval tasks. Future work may explore tighter coupling between knowledge and model training by involving medical knowledge graphs in both pretraining and fine-tuning stages. Such unified frameworks could offer deeper semantic grounding and more robust generalization across diverse clinical retrieval scenarios.

## 4.3 KNOWLEDGE-AUGMENTED VISUAL QUESTION ANSWERING

The knowledge-augmented visual question answering task aims to improve medical visual question answering task (Hasan et al. (2018)) by integrating domain knowledge contained in multimodal medical knowledge graphs, enabling more accurate and clinically meaningful question answering over medical images.

**Datasets and Backbone Models.** To benchmark current knowledge-augmented visual question answering methods with our proposed MEDMKG, we adopt three widely used medical VQA datasets, following previous work (Li et al. (2024)). These datasets include VQA-RAD (Lau et al. (2018)), Slake (Liu et al. (2021)), and Path-VQA (He et al. (2020)). For a fair comparison, we select closed-set questions from the datasets, which can be equally tackled by methods with different sophistication.

Table 4: Results (%) on Medical Visual Question Answering with Knowledge Graphs. Metrics highlighted with green indicate improvement over backbone, while red refers to drop. Augmented by MEDMKG, most of methods achieve better performance on the task, showcasing the usefulness of knowledge condensed in MEDMKG.

| Methods | VQA-RAD | | | | SLAKE | | | | PathVQA | | | |
|---|---|---|---|---|---|---|---|---|---|---|---|---|
| | Acc↑ | Prec↑ | Rec↑ | F1↑ | Acc↑ | Prec↑ | Rec↑ | F1↑ | Acc↑ | Prec↑ | Rec↑ | F1↑ |
| **CLIP** | 64.94 | 62.71 | 62.71 | 62.71 | 65.07 | 62.09 | 74.86 | 67.88 | 81.89 | 88.37 | 76.54 | 82.03 |
| + KRISP | 73.71 | 78.89 | 60.17 | 68.27 | 56.90 | 55.00 | 69.14 | 61.27 | 84.21 | 89.83 | 79.79 | 84.51 |
| + MKBN | 70.12 | 70.87 | 61.86 | 66.06 | 70.14 | 73.47 | 61.71 | 67.08 | 84.68 | 89.35 | 81.33 | 85.15 |
| + K-PathVQA | 66.14 | 62.79 | 68.64 | 65.59 | 69.30 | 73.57 | 58.86 | 65.40 | 84.15 | 85.74 | 84.75 | 85.24 |
| + EKGRL | 67.73 | 65.04 | 67.80 | 66.39 | 70.70 | 71.01 | 68.57 | 69.77 | 84.77 | 86.38 | 85.24 | 85.81 |
| + MR-MKG | 73.71 | 77.08 | 62.71 | 69.16 | 76.34 | 79.74 | 69.71 | 74.39 | 84.30 | 84.85 | 86.34 | 85.59 |
| **PubMedCLIP** | 66.14 | 64.35 | 62.71 | 63.52 | 63.94 | 59.59 | 83.43 | 69.52 | 81.26 | 86.65 | 77.20 | 81.65 |
| + KRISP | 76.10 | 76.85 | 70.34 | 73.45 | 75.77 | 79.47 | 68.57 | 73.62 | 84.41 | 88.13 | 82.21 | 85.07 |
| + MKBN | 67.33 | 67.31 | 59.32 | 63.06 | 70.70 | 75.18 | 60.57 | 67.09 | 84.56 | 90.15 | 80.18 | 84.87 |
| + K-PathVQA | 72.51 | 76.34 | 60.17 | 67.30 | 68.17 | 67.03 | 69.71 | 68.35 | 83.76 | 87.62 | 81.44 | 84.42 |
| + EKGRL | 76.49 | 75.21 | 74.58 | 74.89 | 75.49 | 73.40 | 78.86 | 76.03 | 84.59 | 90.41 | 79.96 | 84.86 |
| + MR-MKG | 78.88 | 76.86 | 78.81 | 77.82 | 77.75 | 78.57 | 75.43 | 76.97 | 84.18 | 86.07 | 84.36 | 85.21 |
| **BioMedCLIP** | 66.93 | 61.74 | 77.97 | 68.91 | 70.14 | 70.18 | 68.57 | 69.36 | 84.56 | 94.03 | 76.27 | 84.22 |
| + KRISP | 76.10 | 79.59 | 66.10 | 72.22 | 57.18 | 54.77 | 75.43 | 63.46 | 85.46 | 93.74 | 78.30 | 85.33 |
| + MKBN | 68.53 | 64.66 | 72.88 | 68.53 | 67.89 | 75.63 | 51.43 | 61.22 | 85.78 | 88.32 | 84.91 | 86.58 |
| + K-PathVQA | 65.34 | 71.23 | 44.07 | 54.45 | 70.70 | 73.20 | 64.00 | 68.29 | 85.93 | 90.57 | 82.54 | 86.37 |
| + EKGRL | 75.70 | 71.76 | 79.66 | 75.50 | 86.20 | 89.38 | 81.71 | 85.37 | 85.46 | 89.66 | 82.60 | 85.98 |
| + MR-MKG | 77.29 | 74.80 | 77.97 | 76.35 | 80.28 | 79.66 | 80.57 | 80.11 | 87.24 | 90.06 | 85.85 | 87.91 |
| **MedCSPCLIP** | 68.13 | 61.59 | 85.59 | 71.63 | 66.20 | 83.95 | 38.86 | 53.12 | 77.72 | 73.37 | 92.24 | 81.73 |
| + KRISP | 80.08 | 84.00 | 71.19 | 77.06 | 70.70 | 91.76 | 44.57 | 60.00 | 83.19 | 94.71 | 72.96 | 82.43 |
| + MKBN | 69.72 | 65.44 | 75.42 | 70.08 | 67.32 | 75.21 | 50.29 | 60.27 | 85.37 | 86.17 | 86.84 | 86.51 |
| + K-PathVQA | 67.73 | 75.34 | 46.61 | 57.59 | 71.55 | 74.03 | 65.14 | 69.30 | 85.31 | 89.35 | 82.65 | 85.87 |
| + EKGRL | 76.10 | 73.39 | 77.12 | 75.21 | 69.30 | 78.95 | 51.43 | 62.28 | 84.92 | 92.75 | 78.19 | 84.85 |
| + MR-MKG | 78.49 | 77.59 | 76.27 | 76.92 | 83.94 | 83.15 | 84.57 | 83.85 | 86.53 | 89.74 | 84.75 | 87.17 |

We use the same set of backbone models as in Section 4.2, namely CLIP (Radford et al. (2021)), PubMedCLIP (Eslami et al. (2023)), BioMedCLIP (Zhang et al. (2023)), and MedCSPCLIP (Wang et al. (2024c)). For more details, please refer to Appendix H.

**Baselines, Evaluation Metrics & Implementation.** We evaluate five models that integrate knowledge graphs to enhance visual question answering: KRISP (Marino et al. (2021)), MKBN (Huang et al. (2023)), K-PathVQA (Naseem et al. (2023)), EKGRL (Ren et al. (2023)), and MR-MKG (Lee et al. (2024)). Detailed descriptions of these approaches are provided in Appendix J.2. We adopt four widely accepted metrics for the visual question answering task: Accuracy, Precision, Recall, and F1 score. More detailed metric descriptions can be found in Appendix J.3. We use the same implementation configuration as described in Section 4.2.

**Evaluation Results.** Table 4 summarizes the performance (%) of knowledge-augmented VQA models across VQA-RAD, SLAKE, and PathVQA. Incorporating external knowledge from our multimodal medical knowledge graph consistently improves model performance, particularly on Accuracy and F1 metrics, confirming the utility of structured domain-specific knowledge in enhancing medical visual reasoning. Among the evaluated methods, MR-MKG achieves the highest and most stable performance across datasets and backbones, underscoring the effectiveness of contrastive learning in promoting robust cross-modal alignment. Attention-based fusion methods (K-PathVQA and MKBN) show less consistent gains, with noticeable performance degradation on smaller datasets (VQA-RAD and SLAKE), likely due to overfitting. However, their improvements stabilize on larger datasets (e.g., PathVQA), suggesting that attention-driven integration requires sufficient data to avoid overfitting to noisy or spurious knowledge signals. In conclusion, the results confirm that incorporating our multimodal medical knowledge graph effectively enhances performance in medical VQA tasks. The graph's clinical specificity, image-aware relational structure, and semantic richness contribute to

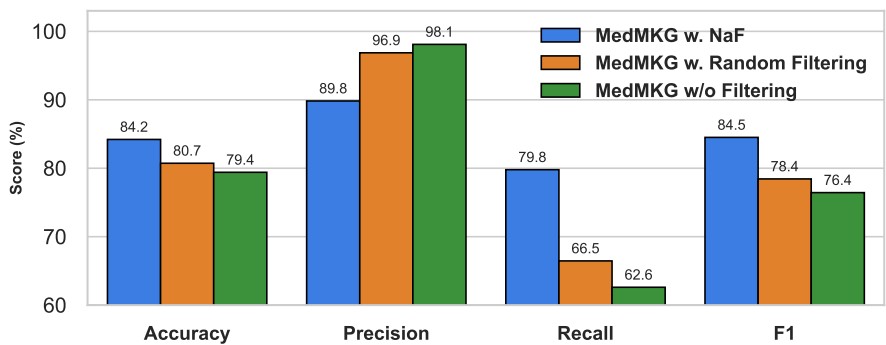

Figure 4: The ablation study on the effectiveness of NaF.

the stronger multimodal understanding. Future work should explore adaptive, backbone-agnostic fusion mechanisms to further improve stability and generalizability across diverse datasets and model architectures.

### 4.4 ABLATION ON NaF

To understand how NaF improves the utility of MEDMKG, we conduct an ablation study using KRISP on the PathVQA dataset with three versions of the graph: (i) the graph filtered by our proposed NaF algorithm, (ii) a graph obtained via random sampling to match NaF's size, and (iii) an unfiltered graph. The results are shown in Figure 4.

The model that relies on the unfiltered graph struggles to extract useful signals due to severe redundancy and noise. As a result, it tends to adopt an overly conservative prediction strategy, yielding high precision but substantially worse recall, and ultimately performs poorly on overall accuracy and F1. Random filtering, by contrast, reduces redundancy and helps the model access more relevant information, but it also removes informative nodes and relations indiscriminately, degrading graph quality and leading to suboptimal performance.

NaF achieves the best results among all three settings. By selectively removing redundant structure while preserving essential graph informativeness by design, NaF provides a cleaner and more discriminative knowledge graph. This confirms NaF's effectiveness in reducing structural redundancy and noise, consistent with our discussion in Section 3.4.

## 5 CONCLUSION

In this work, we present MEDMKG, a novel multimodal medical knowledge graph that integrates clinical text and medical imaging data to capture rich inter- and cross-modality relationships. To ensure the graph's quality and conciseness, we introduce a novel neighbor-aware filtering algorithm tailored to multimodal knowledge graphs. Extensive experiments on knowledge graph analysis and downstream augmentation tasks validate the effectiveness of MEDMKG and highlight its value in enhancing medical knowledge representation. Beyond serving as a valuable resource that can be continuously expanded with data from more diverse patient populations and imaging devices, MEDMKG also opens up new research opportunities. It highlights the need for adaptive and efficient strategies to integrate multimodal knowledge into real-world clinical applications such as report generation, diagnostic reasoning, and temporal prediction.

ETHICS STATEMENT

This work is guided by the principles of contributing to human well-being and avoiding harm. While MEDMKG is intended to advance socially responsible and equitable research, we acknowledge potential risks such as diagnostic errors, biased decision support, or reinforcement of health disparities if models trained on it are misused. To minimize such harms, we stress the importance of expert validation, continuous monitoring of deployed systems, and safeguards against unverified clinical use. We further encourage broad, responsible accessibility of the resource, prioritizing the needs of less advantaged groups and ensuring that its use respects diversity, privacy, and safety across socio-economic contexts.

REPRODUCIBILITY STATEMENT

We are committed to ensuring the reproducibility of our work. A detailed description of the knowledge graph curation process is provided in the Appendix to allow others to replicate the data construction pipeline. The full implementation of our methods and experiments is released at https://anonymous.4open.science/r/MedMKG-525F. To further support replicability, we control for randomness by setting fixed random seeds across all experiments. Together, these efforts provide transparency and enable the community to verify and build upon our results.

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

## A    LLM Usage Statement

In this work, large language models (LLMs) played two complementary roles. First, GPT-4o was directly incorporated into the research pipeline as a tool for biomedical concept extraction and relation identification. These extracted elements served as the basis for constructing and analyzing our knowledge graph, and thus represent an essential component of the technical contributions of this paper. The integration of LLMs into these processes was carefully monitored, and the resulting outputs were cross-checked to ensure alignment with domain knowledge and study objectives.

Second, we employed GPT-4o in a supportive capacity during manuscript preparation. This usage was limited to surface-level improvements such as refining word choice, correcting grammar, and enhancing overall readability. The scientific ideas, experimental design, and interpretations reported in this paper remain entirely those of the authors.

Across both research and writing contexts, all LLM-generated outputs were reviewed for accuracy and appropriateness. The authors take full responsibility for the validity and integrity of the final content.

## B    Compute and Environment Configuration

All experiments were conducted on an NVIDIA A100 GPU with CUDA version 12.0, running on an Ubuntu 20.04.6 LTS server.

## C    Dataset Repository

We have provided a anonymous dataset repository for MEDMKG, available at https://anonymous.4open.science/r/MedMKG-525F. The MEDMKG dataset can be loaded alongside the MIMIC-CXR dataset, which requires separate download following the instructions provided in the repository README file. The repository also includes runnable code for data processing, baseline models, environment configuration, and example execution scripts. We are committed to publication of the repository after the acceptance of this study, as well as regularly updating the repository with additional modalities, datasets, and tasks to further support the research community.

## D    Deployment and Updating

MEDMKGsupports three categories of updates:

1. **Foundational knowledge updates** (e.g., incorporating new UMLS releases).
2. **Imaging dataset updates** (e.g., newly added MIMIC–CXR studies or revised radiology reports).
3. **Multimodal extensions** (e.g., integration of CT, MRI, ultrasound, or EHR-derived features).

Because the construction pipeline is highly efficient, typically requiring only a few hours with API-based processing, the entire workflow can be re-executed whenever new data or modalities become available, thereby enabling continuous and real-time maintenance of MEDMKG.

## E    Details of Knowledge Graph Construction

### E.1    Pre-processing of MIMIC-CXR

To ensure the quality of the constructed multimodal knowledge graph, we perform targeted pre-processing on the raw data in the MIMIC-CXR database. Each radiological report may correspond to images in different views, including anteroposterior, posteroanterior, lateral, etc. Involving multiple images with the same set of concepts could result in significant redundant edges within the knowledge graph. Therefore, we only maintain images in the anteroposterior view for graph conciseness;

Table 5: Filtered Semantic Types. The semantic types listed below are disallowed; all others are considered allowable.

| | | |
|---|---|---|
| Occupation or Discipline | Intellectual Product | Age Group |
| Biomedical Occupation or Discipline | Classification | Patient or Disabled Group |
| Organization | Regulation or Law | Geographic Area |
| Health Care Related Organization | Language | Conceptual Entity |
| Professional Society | Group Attribute | Idea or Concept |
| Self-help or Relief Organization | Group | Temporal Concept |
| Professional or Occupational Group | Qualitative Concept | Quantitative Concept |
| Population Group | Functional Concept | Body System |
| Family Group | | |

similarly, radiological reports usually contain abundant information such as diagnostic history that is not directly relevant to the content of the corresponding radiological image, therefore, extracting concepts from these similar reports can also result in redundancy.

To mitigate this problem, we only preserve sections of Impression and Findings, two major sections that contain the most informative content, and stick to existing works in clinical report analysis (Luo et al. (2024)). We perform semantic filtering using DBSCAN (Ester et al. (1996)) and MedCSP-CLIP (Wang et al. (2024c)). To be specific, we encode all the radiological reports with the text encoder of MedCSPCLIP, then perform clustering on the reports based on their semantics. Based on the clustering results, we select the ones near the centroid of each cluster as representative of a group of similar radiological reports.

These approaches function together, ensuring that our pipeline referred to in Section 3 receives high-quality data for processing, producing the multimodal knowledge graph with sufficient information, negligible noise, and minimal redundancy.

### E.2 FILTERING PER SEMANTIC TYPE OF MEDICAL CONCEPTS

In order to eliminate concepts that are overly abstract or lack practical value, we filter concepts based on their semantic types. Table 5 lists the semantic types that are not preferred thus filtered, while all other semantic types in the UMLS vocabulary [1] are allowed.

### E.3 PROMPT FOR CONCEPT DISAMBIGUATION AND RELATION EXTRACTION

To leverage the LLM's contextual understanding for effective concept disambiguation and relation extraction, we designed an instructive prompt that guides the model through these tasks. The prompt is presented in Example E.3.

### E.4 SELECTION OF LLM

In this study, GPT-4o (OpenAI Achiam et al. (2023)) is selected for disambiguation, as prior research has demonstrated its superior performance in biomedical comprehension (Silberg et al. (2024); Dataset) and its effectiveness in resolving medical terminology ambiguity (Kugic et al. (2024)), compared with other LLMs. To further substantiate this choice, we conduct a case study to evaluate the suitability of GPT-4o for curating MEDMKG, as shown in Example E.4.

The analysis reveals that GPT-4o outperforms other advanced LLM backbones, i.e., Gemini-2.5 (Comanici et al. (2025)) and LLaMA-3.1-8B-Instruct (Grattafiori et al. (2024)), by extracting more accurate concepts of interest while generating fewer hallucinations. These findings reinforce our decision to adopt GPT-4o for concept and relation extraction.

---

[1] https://www.nlm.nih.gov/research/umls/META3_current_semantic_types.html

---

**Prompt for Concept Disambiguation and Relation Extraction (E.3)**

Report Text: [*Report Text*]
Candidate Concepts: [*Candidate Concepts*]
For each phrase, evaluate the concept candidates and select the most relevant concept based on the context provided in the report. Your decision should account for the specific context of a radiological image.
After selecting the appropriate concept for each phrase, classify the relation between the selected concept and the image using the following categories:
**Positive** - The concept is clearly represented in the image (e.g., anatomical structures, specific findings).
**Neutral** - Concepts that are structural, general terms (like "findings", "normal", "changes"), meta-concepts, adjectives, or unrelated to clinical insight.
**Negative** - The concept is the opposite of what is shown in the image (e.g., when the image shows no abnormalities but the concept implies pathology).
**Uncertain** - The concept's presence in the image is unclear based on the report (e.g., the reporter uses language like "possible" or "could be").
Return only concepts with a positive, negative, or uncertain relation. Do not include any neutral concepts in the final output.
Provide the final output in the following format: ***start***
(Concept ID only (digits start with C), Relation)
***end***
Ensure that:

- Neutral concepts are excluded entirely from the output.

- Concepts like "findings" and any general or structural terms are categorized as neutral and omitted.

- Double-check that each remaining concept is evaluated accurately based on the context of the radiological image.

---

## E.5 NAF ALGORITHM

We propose the Neighbor-Aware Filtering (NaF) algorithm for effective image filtering to boost the conciseness of MEDMKG. More details are presented in Algorithm 1.

## E.6 ILLUTRATION OF MEDMKG

Figure 5 shows a subgraph of MEDMKG, provided to facilitate a better understanding of its structure and content. As shown in Figure 5, the medical multimodal knowledge graph integrates both intra- and cross-modal edges, offering rich multimodal medical knowledge that can potentially support a wide range of applications.

# F DETAILS OF HUMAN ASSESSMENT

## F.1 ASSESSMENT CRITERIA

We conducted a human evaluation to assess the quality of MEDMKG. Three key metrics were used:

- **Concept Coverage** measures how comprehensively the extracted concepts capture the clinically meaningful findings present in the image.

- **Relation Correctness** assesses whether the relationships between images and extracted concepts are accurately modeled, correctly identified with positive, negative, or uncertain associations.

- **Image Diversity** evaluates whether the set of images associated with each concept reflects a diverse range of clinical scenarios, rather than highly homogeneous ones.

---

**Algorithm 1** Neighbor-Aware Filtering Algorithm

---

1: **Input:**
   - A set of images $\mathcal{M} = \{m_1, m_2, \ldots, m_N\}$.
   - For each image $m_i$, its associated triplets $T_i = \{(m_i, r_{ij}, c_{ij})\}$.
   - The set of filtered clinical concepts $\mathcal{C}$.
2: **Output:** Selected image set $\mathcal{M}^*$.
3: $\mathcal{M}^* \leftarrow \emptyset$ and $\mathcal{C}^* \leftarrow \emptyset$.
4: **for** each image $m_i \in \mathcal{M}$ **do**
5:     Compute $\text{Score}(m_i) \leftarrow \sum_{(r,c) \in T_i} \log \dfrac{N}{N_{(r,c)}}$.
6: **end for**
7: Sort $\mathcal{M}$ in descending order by $\text{Score}(m_i)$.
8: **for** each image $m_i$ in sorted order **do**
9:     **if** $\mathcal{C}^* \neq \mathcal{C}$ **then**
10:         $\mathcal{M}^* \leftarrow \mathcal{M}^* \cup \{m_i\}$.
11:         $\mathcal{C}^* \leftarrow \mathcal{C}^* \cup \{c \mid \exists r \text{ such that } (r,c) \in T_i\}$.
12:     **else**
13:         **break**
14:     **end if**
15: **end for**
16: **return** $\mathcal{M}^*$.

---

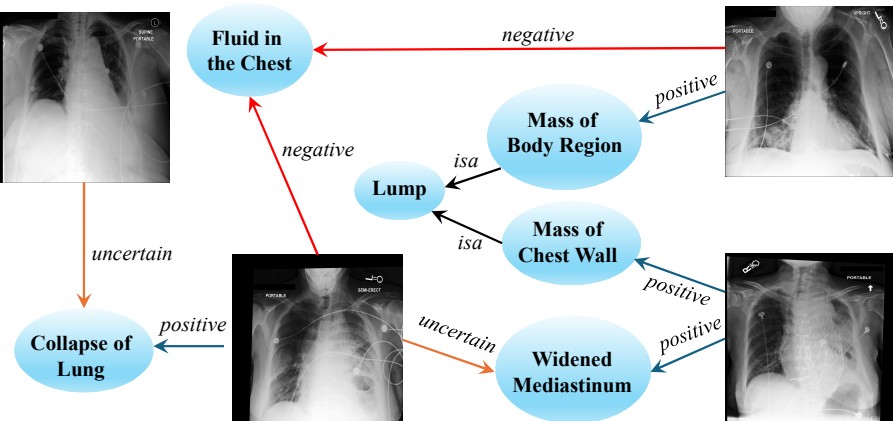

Figure 5: An illustration of MEDMKG.

These metrics were selected to capture complementary aspects of performance: *Concept Coverage* ensures clinical relevance and completeness; *Relation Correctness* ensures accurate representation of image-concept associations; and *Image Diversity:* ensures the robustness and generalizability of concept representations. Together, they provide a holistic evaluation of both precision and breadth of MEDMKG.

## F.2 ASSESSMENT PROCEDURE

For the metrics of concept coverage and relation correctness, we randomly sample 30 images in MEDMKG, choose all their concept neighbors, and the relation connecting them for assessment. For image diversity, we randomly choose 30 concepts in MEDMKG and provide all the images positively linked with them to the evaluator. The evaluator performs the assessment along with detailed guidance.

---

**Entity & Relation Extraction Comparison Across LLMs (E.4)**

**Radiological Report:**
*As compared to the previous radiograph, there is a further increase in extent of the opacities in the right lung. The left lung is constant. Changed nasogastric tube. Moderate cardiomegaly with extensive retrocardiac atelectasis.*

**Results:**

**Gemini-2.5**

```
C0032285:Pneumonia, Uncertain
C0225706:Right Lung, Positive
C0029053:opacities, Positive
C0225730:Left Lung, Positive
C0018800:Cardiomegaly, Positive
C0004144:Atelectasis, Positive
```

**LLaMA-3.1**

```
C0032285:PNEUMONIA (Pneumonia), Positive
C0264716:Chronic heart failure, Positive
C0476273:Distress, Respiratory (Respiratory distress),
Positive
```

**GPT-4o**

```
C0029053:opacities (Decreased translucency), Positive
C0225706:Right Lung (Right lung), Positive
C0085678:Nasogastric Tube (Nasogastric tube), Positive
C0018800:CARDIOMEGALY (Cardiomegaly), Positive
C0004144:ATELECTASIS (Atelectasis), Positive
```

---

## G  DETAILS OF LINK PREDICTION

### G.1  LINK PREDICTION BASELINES

We benchmark MEDMKG with the following baseline models in the task of link prediction:

- **AttH** (Chami et al. (2020)) is a hyperbolic knowledge graph embedding model designed to capture hierarchical structures by leveraging the Lorentz model.

- **DistMult** (Yang et al. (2014)) is a bilinear factorization model for knowledge graphs that represents relations as diagonal matrices, enabling efficient computation.

- **TransR** (Lin et al. (2015)) extends TransE by introducing separate relation-specific entity spaces, allowing better modeling of diverse relationships.

- **HypER** (Balažević et al. (2019a)) applies hypernetworks to generate relation-dependent transformation matrices for entity embeddings, improving flexibility.

- **SimplE** (Kazemi & Poole (2018)) is an extension of Canonical Polyadic (CP) decomposition that enables each entity representation to be used in two different ways.

- **TuckER** (Balažević et al. (2019b)) is based on Tucker decomposition and factorizes the knowledge graph tensor into entity and relation embeddings with a core interaction tensor.

- **MurP** (Balazevic et al. (2019)) embeds knowledge graphs in the Poincaré ball model, enabling effective representation of hierarchical data.

- **MurE** (Balazevic et al. (2019)) embeds knowledge graphs in Euclidean space using multiple relational constraints to improve predictive performance.
- **NTN** (Socher et al. (2013)) introduces a neural tensor network for knowledge graph embedding, modeling entity interactions through a bilinear tensor layer.
- **TransD** (Ji et al. (2015)) extends TransE and TransH by introducing entity- and relation-specific projection matrices for dynamic embedding transformation.
- **TransE** (Bordes et al. (2013)) models relationships as translations in the embedding space, assuming that the sum of the head and relation embeddings approximates the tail embedding.
- **RESCAL** (Nickel et al. (2011)) models multi-relational data using a bilinear tensor factorization approach that captures pairwise interactions.
- **RotatE** (Sun et al. (2019)) represents relations as rotations in a complex vector space, capturing symmetric and antisymmetric relations effectively.
- **TransH** (Wang et al. (2014)) introduces relation-specific hyperplanes to improve the representation of diverse relational properties.
- **ConvE** (Dettmers et al. (2018)) applies 2D convolutional neural networks to entity embeddings, capturing complex interactions between entities and relations.
- **ComplEx** (Trouillon et al. (2016)) extends DistMult by using complex-valued embeddings, enabling the representation of asymmetric relations.
- **ConvR** (Jiang et al. (2019)) applies relation-specific convolutional filters to entity embeddings, enhancing the modeling of complex interactions.

## G.2 EVALUATION METRICS

For the link prediction tasks, we utilize Mean Rank (MR) and Hits@K for assessing the baselines. Let $\mathcal{T}$ denote the set of test triples and, for each test case $i$, let $r_i$ be the rank of the ground-truth entity among all candidate entities (with a lower rank indicating better performance). The metrics are defined as follows:

**Mean Rank (MR)** The Mean Rank is the average rank of the ground-truth entities over all test cases:

$$\text{MR} = \frac{1}{|\mathcal{T}|} \sum_{i=1}^{|\mathcal{T}|} r_i. \tag{2}$$

**Hits@K** Hits@K measures the proportion of test cases for which the ground-truth entity is ranked within the top $K$ predictions:

$$\text{Hits@}K = \frac{1}{|\mathcal{T}|} \sum_{i=1}^{|\mathcal{T}|} \mathbb{I}(r_i \leq K), \tag{3}$$

where $\mathbb{I}(\cdot)$ is the indicator function that returns 1 if the condition is true and 0 otherwise.

A lower MR and a higher MRR or Hits@K value indicate better performance.

## H BACKBONE MODELS IN KNOWLEDGE-AUGMENTED TASKS

The following advanced visual language models are adapted as the standard backbone for knowledge-augmented methods:

- **CLIP** (Radford et al. (2021)) is a vision-language model trained on large-scale internet data using contrastive learning. It aligns images and text embeddings in a shared latent space, enabling zero-shot image classification and retrieval. The model is under the MIT License.
- **PubmedCLIP** (Eslami et al. (2023)) is a domain-specific adaptation of CLIP trained on PubMed articles and biomedical images. It enhances the alignment of biomedical images with textual descriptions, improving zero-shot performance in medical imaging tasks. The model is under the MIT License.

- **BioMedCLIP** (Zhang et al. (2023)) is a biomedical contrastive pretraining model trained on a large-scale corpus of biomedical images and text. It is designed to improve multimodal understanding in healthcare applications, particularly for retrieval and classification tasks. The model is under the MIT License.
- **MedCSPCLIP** (Wang et al. (2024c)) is a medical-specific adaptation of CLIP that incorporates the MedCSP framework for contrastive scalable pretraining. It learns generalizable medical image representations, enabling improved zero-shot performance and transfer learning in clinical applications. The model is under the MIT License.

## I  DETAILS OF KNOWLEDGE-AUGMENTED IMAGE-TEXT RETRIEVAL

### I.1  BASELINES

In the task of knowledge-augmented image-text retrieval, we benchmark with the following baseline models:

- **KnowledgeCLIP (Pan et al. (2022))**: This model extends CLIP by integrating external knowledge graphs. By adding knowledge-based objectives during pre-training, it leverages structured relational data (e.g., from ConceptNet or VisualGenome) to improve semantic alignment between images and text.
- **FashionKLIP (Wang et al. (2023b))**: Designed for the fashion domain, FashionKLIP automatically constructs a multimodal conceptual knowledge graph (FashionMMKG) from large-scale fashion data. By injecting domain-specific knowledge into the pre-training process, it learns fine-grained representations that enhance image-text alignment and retrieval performance.

### I.2  EVALUATION METRICS

For this task, we leverage Precision k and Recall k as the metrics for evaluation. Let $\mathcal{Q}$ denote the set of queries. For each query $q \in \mathcal{Q}$, let $R(q)$ be the set of relevant items, and let $\hat{R}_k(q)$ be the set of top-$k$ items retrieved by the model. Then, the metrics are defined as follows:

**Precision k**  Precision k is the fraction of the top-$k$ retrieved items that are relevant. Formally, it is given by:

$$\text{Precision@}k = \frac{1}{|\mathcal{Q}|} \sum_{q \in \mathcal{Q}} \frac{|\hat{R}_k(q) \cap R(q)|}{k}. \tag{4}$$

**Recall k**  Recall k is the fraction of the relevant items that are retrieved in the top-$k$ results. It is defined as:

$$\text{Recall@}k = \frac{1}{|\mathcal{Q}|} \sum_{q \in \mathcal{Q}} \frac{|\hat{R}_k(q) \cap R(q)|}{|R(q)|}. \tag{5}$$

A higher Precision k indicates that a larger proportion of the retrieved items are relevant, whereas a higher Recall k suggests that a greater proportion of all relevant items have been retrieved. These metrics together provide a comprehensive evaluation of the retrieval performance.

## J  DETAILS OF KNOWLEDGE-AUGMENTED VISUAL QUESTION ANSWERING

### J.1  DATASETS

We compare the baselines on three medical visual question answering dataset, including VQA-RAD, SLAKE and PathVQA. We extract closed questions in these datasets for benchmarking.

### J.2  BASELINES

In the task of knowledge-augmented visual question answering, we evaluate five models that incorporate external knowledge graphs to improve visual reasoning and answer prediction:

- **KRISP (Marino et al. (2021))**: This model integrates structured knowledge graphs into the VQA pipeline, refining both image representations and question understanding to boost answer accuracy.

- **MKBN (Huang et al. (2023))**: Originally designed for medical VQA, MKBN leverages domain-specific knowledge graphs to align visual and textual features, thus enhancing performance in specialized settings.

- **K-PathVQA (Naseem et al. (2023))**: By incorporating multi-hop reasoning over a knowledge graph, K-PathVQA enables the model to infer complex relationships and answer questions that require multi-step deductions.

- **EKGRL (Ren et al. (2023))**: This framework combines graph-based representation learning with reinforcement learning to effectively integrate external knowledge, thereby improving reasoning capabilities in visual question answering.

- **MR-MKG (Lee et al. (2024))**: MR-MKG utilizes contrastive loss to capture diverse semantic interactions between visual content and questions, leading to enhanced cross-modal alignment and VQA performance.

### J.3 EVALUATION METRICS

For the visual question answering task, we adopt four standard metrics: Accuracy, Precision, Recall, and F1 score. Let $\mathcal{D}$ denote the set of VQA examples. For each example $i \in \mathcal{D}$, let $y_i$ be the ground-truth answer and $\hat{y}_i$ the predicted answer. The metrics are defined as follows:

**Accuracy**    Accuracy measures the proportion of correctly answered questions:

$$\text{Accuracy} = \frac{1}{|\mathcal{D}|} \sum_{i \in \mathcal{D}} \mathbb{I}(\hat{y}_i = y_i), \tag{6}$$

where $\mathbb{I}(\cdot)$ is the indicator function.

**Precision**    Precision is the fraction of true positive answers among all answers predicted as positive. In a binary (or thresholded) setting, it is given by:

$$\text{Precision} = \frac{TP}{TP + FP}, \tag{7}$$

with $TP$ and $FP$ denoting the numbers of true positives and false positives, respectively.

**Recall**    Recall is defined as the fraction of true positive answers among all actual positive answers:

$$\text{Recall} = \frac{TP}{TP + FN}, \tag{8}$$

where $FN$ represents false negatives.

**F1 Score**    The F1 score is the harmonic mean of Precision and Recall:

$$\text{F1} = 2 \cdot \frac{\text{Precision} \cdot \text{Recall}}{\text{Precision} + \text{Recall}}. \tag{9}$$

Together, these metrics provide a comprehensive evaluation of model performance on the knowledge-augmented visual question answering task.

