# OpenReview forum: "MEDMKG: Benchmarking Medical Knowledge Exploitation with Multimodal Knowledge Graph"
_ICLR.cc/2026/Conference — ICLR 2026 Conference Withdrawn Submission_

### Official Review · Reviewer_qVVF · 2025-10-28

**Soundness:** 3
**Presentation:** 3
**Contribution:** 3
**Rating:** 6
**Confidence:** 3

**Summary:**

This paper addresses the problem of existing medical deep learning models relying on single-modal knowledge graphs and lacking resources linking visual and clinical concepts. It proposes a medical multimodal knowledge graph, MEDMKG, through a multi-stage construction process that fuses multimodal data from MIMIC-CXR with structured clinical knowledge from UMLS. It introduces a Neighbor-aware Filtering (NaF) algorithm to ensure graph quality and simplicity. The paper benchmarks 24 baseline methods and four mainstream vision-language backbone models across five tasks (link prediction, text-image retrieval, and visual question answering) under two experimental settings. Results demonstrate that MEDMKG not only improves the performance of downstream medical tasks but also provides a foundation for the development of adaptive and robust strategies for integrating multimodal knowledge in medical AI. The paper also identifies key insights, such as the need for model selection to match graph structure and the potential for external knowledge to improve downstream task performance.

**Strengths:**

1. It breaks the limitations of traditional unimodal medical knowledge graphs by innovatively integrating MIMIC-CXR imaging data with UMLS clinical knowledge to construct MEDMKG, the first multimodal knowledge graph tailored for medical scenarios. The proposed NaF algorithm offers a new approach to redundancy filtering in multimodal knowledge graphs by balancing image connectivity and uniqueness.

2.The construction process is rigorous, combining MetaMap and GPT-4o to achieve high-precision concept extraction and relationship modeling. The graph's quality is validated through expert evaluations and quantitative analyses. The experimental design is comprehensive, covering 5 tasks, 2 experimental settings, and 24 baseline methods, ensuring reliable and statistically meaningful results.

3.The paper follows a standardized structure with a logical flow from problem formulation to solution development and experimental validation. Key technologies and experimental details are described in detail, facilitating reproduction and reference by other researchers.

4.It fills the gap in resources for multimodal medical knowledge graphs. Experimental results confirm that MEDMKG can effectively improve the performance of downstream medical tasks, providing a new paradigm for multimodal knowledge integration in medical AI and making a significant contribution to advancing the field of medical intelligence.

**Weaknesses:**

1.The core parameters of the NaF algorithm (e.g., the determination method of M in the formula) are not elaborated, and no parameter sensitivity analysis is conducted to verify the impact of different parameter settings on graph quality and downstream task performance. Furthermore, there is no comparison with other mainstream knowledge graph filtering algorithms, making it difficult to highlight the advantages of NaF.

2.Experimental data mainly rely on MIMIC-CXR (chest X-ray images) and UMLS, without extension to other medical imaging types (e.g., CT, MRI) or clinical datasets (e.g., electronic health records, EHR). This fails to fully validate the generalization ability of MEDMKG across multiple medical scenarios. In some tasks, the performance of head entity prediction is significantly lower than that of tail entity prediction, but the fundamental cause of this difference is not analyzed in depth.

3.The paper does not discuss the feasibility of deploying MEDMKG in real clinical scenarios, such as the graph's update mechanism (how to integrate new medical knowledge and imaging data), compatibility with existing clinical systems, and specific strategies to address potential ethical risks (e.g., diagnostic bias) in clinical decision support.

**Questions:**

1.In the NaF algorithm, is M (the total number of medical images in the knowledge graph) a fixed value or dynamically adjusted based on the dataset? If dynamically adjusted, what is the basis for adjustment? Can you supplement comparative experiments on algorithm performance under different M values to verify the rationality of parameter selection?

2.In the experiment, the performance of head entity prediction is lower than that of tail entity prediction. The authors attribute this to modal heterogeneity. Can you further analyze the specific differences between image and text modalities in the embedding space and propose targeted optimization directions?

3.Currently, MEDMKG only integrates chest X-ray images and UMLS knowledge. How do you plan to extend it to other medical imaging types (e.g., CT, ultrasound) or clinical data (e.g., laboratory test results) in the future? During the extension process, how will you resolve semantic inconsistency issues among different modal data?

4.The paper mentions that MEDMKG can be used for clinical decision support. Can you elaborate on its deployment process in real clinical scenarios? For example, how to interface with existing hospital systems, how to ensure the real-time and accuracy of graph updates, and how to avoid potential diagnostic bias risks?

---

> ### Author Response · Authors · 2025-11-26
> **Response to Reviewer qVVF (1)**
>
> We thank the reviewer for the insightful feedback. Below we address each weakness and question point by point.
>
> ---
>
> ## W1. Missing parameter sensitivity analysis for NaF / unclear definition of M
>
> As detailed in Line 255 of the paper, **M represents the total number of medical images in the *pre-filtered* knowledge graph**. Similar to other mathematical quantities in Eq. (1), **M is not a tunable hyperparameter**, nor is it intended to be adjusted during the algorithm. It is simply a **statistic of the graph structure itself**.
>
> Thus, for any given pre-filtered graph, NaF is **fully deterministic** and does not involve any parametric sensitivity that could influence graph quality or downstream task performance. We will clarify this more explicitly in the revision to avoid misunderstanding.
>
> ---
>
> ## W2. Reliance on MIMIC-CXR and UMLS / performance gap in head vs. tail prediction
>
> We agree that the current version of MEDMKG uses only chest X-rays and UMLS knowledge. As noted, the **construction method is fully generalizable** and can be applied to any multimodal dataset where image–text (or modality–text) correspondence exists (e.g., CT–report, ultrasound–report, MRI–report). Extending to broader datasets is a major goal of our future work.
>
> Regarding head vs. tail performance differences:
>
> - **Image and concept nodes participate differently in the graph.**
>   - Images connect *only* to concepts.
>   - Concepts connect to *both* images and text-level relations.
> - This heterogeneous connectivity makes the **shared semantic embedding space more complex**, and head prediction (which may involve either images or text nodes) therefore has a **much larger search space** than tail prediction.
> - Standard link prediction methods treat head and tail symmetrically, which is not ideal for multimodal graphs.
>
> We have revised Section 4.1 and suggest a **a modality-aware link prediction module** that adaptively handles images and text differently, which is an important direction for future improvement.
>
> ---
>
> ## W3. Lack of discussion on deployment, updating, and clinical feasibility
>
> MEDMKG can be updated in three ways:
>
> 1. **Updates to foundational medical knowledge** (e.g., new UMLS releases).
> 2. **Updates to imaging datasets** (e.g., new MIMIC-CXR studies, updated radiology reports).
> 3. **Integration of new modalities** (e.g., CT, MRI, ultrasound, EHR features).
>
> Because the construction pipeline is highly efficient (hours with API processing), the entire workflow can be **re-run whenever new data or modalities are added**, enabling real-time maintenance. We have added Appendix D for this claim.
>
> Regarding clinical deployment, we have already taken steps to mitigate ethical risks (as noted in the ethics statement). We will further refine this in future work through:
>
> - expert-in-the-loop validation,
> - iterative feedback cycles with clinical practitioners, and
> - monitoring mechanisms to detect biases or unintended behaviors.
>
> We have added the discussion in Appendix D for clarification.

---

> ### Author Response · Authors · 2025-11-26
> **Response to Reviewer qVVF (2)**
>
> ## Q1. Is M fixed or dynamically adjusted? Should there be experiments for different M values?
>
> As noted in W1, **M is fixed for any given pre-filtered graph**.
> It is not a tunable parameter and does not vary with datasets or hyperparameters. Therefore, sensitivity analysis is not applicable.
>
> ---
>
> ## Q2. Why is head prediction weaker? Can you analyze modality-specific differences?
>
> Yes. The performance gap arises from **modal heterogeneity**:
>
> - Image nodes have only image→concept edges.
> - Concept nodes have both textual and visual connections.
> - Their embedding distributions differ due to their inherently different structures.
>
> Thus, head prediction, which may involve predicting an image node within a larger, more heterogeneous search space, is more challenging.
>
> We have expanded this explanation and propose **modality-aware link prediction** as a promising optimization direction.
>
> ---
>
> ## Q3. How will MEDMKG be extended to other modalities or clinical data? How to resolve semantic inconsistency?
>
> Extension is straightforward:
>
> - any modality paired with corresponding reports can be processed,
> - the same pipeline (MetaMap → LLM disambiguation → NaF filtering) applies unchanged, and
> - modality-specific nodes can be incorporated without structural modification.
>
> Semantic inconsistencies (e.g., differences in lexicon across modalities) can be mitigated through:
>
> - UMLS normalization,
> - LLM-assisted synonym/hypernym mapping,
>
> as what we proposed in Section 3.2 and 3.3.
>
> ---
>
> ## Q4. Deployment feasibility and avoiding diagnostic bias?
>
> Deployment could proceed via:
>
> - **integration with hospital information systems** through existing HL7/FHIR-compatible interfaces,
> - **periodic graph reconstruction** to ensure real-time knowledge updates,
> - and **expert review mechanisms** to detect and mitigate diagnostic bias.
>
> As noted in W3, our pipeline supports efficient re-building, enabling real-time updates. Ethical considerations are explicitly highlighted in the paper, and future work will involve deeper collaboration with domain experts for monitoring and risk mitigation.
>
> ---
>
>
> Again, we thank the reviewer again for the thorough feedback.

---

> ### Comment · Reviewer_qVVF · 2025-11-26
>
> Thank you for the author's detailed response, which basically resolved my questions

---

> > ### Author Response · Authors · 2025-11-26
> >
> > Thanks for acknowledging our effort, and we are glad to see that our response successfully addresses your concern.

---

### Official Review · Reviewer_GHE9 · 2025-10-28

**Soundness:** 2
**Presentation:** 2
**Contribution:** 2
**Rating:** 2
**Confidence:** 3

**Summary:**

This paper introduces MEDMKG, a Medical Multimodal Knowledge Graph that integrates textual concepts from UMLS with radiological images from MIMIC-CXR. The authors design a multi-stage construction pipeline combining rule-based extraction (MetaMap) and LLM-based disambiguation (GPT-4o), and propose a simple Neighbor-aware Filtering (NaF) heuristic to select informative images. They benchmark the resulting graph across two settings: (1) link prediction and (2) downstream multimodal tasks (text-image retrieval and VQA) using four vision-language backbones. The results show moderate but consistent performance gains when external knowledge from MEDMKG is integrated.

**Strengths:**

1. Comprehensive baseline coverage for link prediction tasks.
2. Integrating structured clinical knowledge with medical imaging is quite novel in the clinical ML domain.

**Weaknesses:**

W1. The paper claims that multimodal KGs outperform text-only KGs, but no direct comparison is provided. Without a UMLS-only baseline, the improvement cannot be attributed to multimodality itself, leaving the core hypothesis untested.

W2. The proposed Neighbor-aware Filtering (NaF) is a handcrafted scoring rule combining neighbor count and distinctiveness. It is not compared to simpler alternatives such as random sampling, degree-based filtering, or embedding-based diversity selection. Without such comparisons, it is unclear whether NaF truly improves graph quality.

W3. The observed improvements in retrieval and VQA could stem from larger data volume or overlap between MIMIC-CXR and evaluation datasets, rather than genuine multimodal reasoning. An ablation that isolates the contribution of image-concept links versus pure text-based knowledge would be essential to establish causal validity

W4. Experiments are reported with single runs and no variance estimates. Results should be averaged across 3-5 random seeds with significance testing to confirm the robustness of claimed improvements.

W5. A case study or visualization showing how MEDMKG influences predictions (e.g., specific examples of correctly or incorrectly augmented VQA answers) would greatly improve transparency.

**Questions:**

If the author could help clarify the following questions, I am happy to raise the score.
1. How does MEDMKG compare to using UMLS alone as a knowledge graph?
2. Can you provide quantitative ablation showing the contribution of the NaF step versus simple baselines?
3. I don't understand the description of dataset leakage prevention. (L374-376) "To prevent any potential data leakage regarding MIMIC-CXR, we only select text-image pairs that were not used during the curation of MEDMKG, and we randomly sample a fixed set of 10,000 pairs from these remaining examples." -- Why were some text-image pairs not used during the curation of MEDMKG? Does that mean MEDMKG includes only a representative subset of text-image pairs?
4. How sensitive are the results of Table 3 and Table 4 for different random seeds?
5. Could you include error analysis to show which question types or retrieval categories benefit most from the multimodal graph?
6. When I click on the anonymized GitHub link in the appendix, the code and dataset are not found. Is there are technical issue?

---

> ### Author Response · Authors · 2025-11-26
> **Response to Reviewer GHE9**
>
> We sincerely thank the reviewer for the detailed assessment and constructive feedback. Below we address weaknesses and questions below:
>
>
> # **W1 & Q1. Lack of comparison against UMLS-only KG**
>
> We appreciate the reviewer’s emphasis on understanding the contribution of multimodality. Conceptually, comparing MEDMKG to a UMLS-only KG is valuable. However, the multimodal baselines used in our evaluation require multimodal structure consisting of:
>
> - visual nodes
> - image–concept links
> - multimodal neighborhoods for grounding
>
> A unimodal KG (UMLS-only) **cannot supply these components**, and as a result, these multimodal architectures cannot meaningfully operate on or benefit from a text-only KG.
>
>
> ---
>
> # **W2 & Q2. NaF not compared to simpler alternatives**
>
> We agree that demonstrating the added value of the Neighbor-aware Filtering (NaF) step is important. In response, we conducted an **additional ablation study (Section 4.4)** comparing:
>
> - MEDMKG w/ NaF
> - MEDMKG w/ Random Filtering
> - MEDMKG w/o Filtering
>
> Results show that:
>
> - **No filtering** leads to dense, noisy neighborhoods that harm the performance of baseline due to redundancy.
> - **Random filtering** reduces graph size but degrades semantics by removing informative nodes.
> - **NaF**, by combining local neighbor count and distinctiveness, preserves informativeness while reducing redundancy, resulting in the best downstream performance.
>
>
> ---
>
> # **W3. Possible overlap between MEDMKG and downstream datasets**
>
> Thank you for raising this concern. We confirm that **there is no overlap** between the subset of MIMIC-CXR used during MEDMKG construction and the samples used for downstream evaluation.
>
> Specifically:
>
> - During curation, NaF filtering removes a portion of image–text pairs from the raw corpus.
> - **MEDMKG includes only the filtered subset**, not the full MIMIC-CXR dataset.
> - For retrieval experiments, we **only sample from the remaining pairs excluded from MEDMKG**.
>
> Thus, improved performance cannot originate from data overlap but rather reflects the structured multimodal information encoded in MEDMKG. We will clarify this point in the paper.
>
> ---
>
> # **W4 & Q4. Missing variance estimates**
>
> We agree that robustness is essential. We have updated **Table 2** to report mean ± standard deviation over **5 random seeds**. Due to computational cost, we are still running multi-seed experiments for Tables 3 and 4, and we will include all results in the camera-ready version.
>
> ---
>
> # **W5 & Q5. Need for case study and error analysis**
>
> We appreciate the suggestion to improve transparency. We are working with domain-expert to prepare the qualitative case study illustrating how MEDMKG changes retrieval/VQA outcomes by benefitting from the clinical knowledge, and will involve it in the final version.
>
> ---
>
> # **Q3. Clarification on leakage prevention**
>
> Thanks for seeking clarification. To elaborate:
>
> - MEDMKG is built using a **filtered subset** of the MIMIC-CXR image–text pairs.
> - A large number of samples are **not selected** because NaF removes noisy or redundant paths.
> - For downstream evaluation, we draw exclusively from the **remaining pairs excluded** by NaF.
>
> Thus, no sample used in the KG appears in the downstream test sets.
>
> ---
>
> # **Q6. GitHub link accessibility**
>
> We checked the anonymized repository (https://anonymous.4open.science/r/MedMKG-525F) and confirmed that code and dataset instructions are accessible. If the reviewer encountered technical issues, this may be related to temporary availability on 4open. After acceptance, we will provide a **public, non-anonymized GitHub repository** with datasets and experiment scripts.
>
> ---
>
>
> We thank the reviewer again for the insightful feedback. These comments along with revision meaningfully strengthen the rigor and clarity of the paper, and we appreciate the reviewer’s guidance in improving the work.

---

### Official Review · Reviewer_2Vrj · 2025-10-29

**Soundness:** 2
**Presentation:** 3
**Contribution:** 2
**Rating:** 2
**Confidence:** 5

**Summary:**

This paper introduces MEDMKG, a medical multimodal knowledge graph built by linking chest X-ray images, their paired radiology reports, and UMLS clinical concepts. The authors describe a pipeline to extract clinical findings from reports, align them to ontology terms, associate them with images, filter the resulting graph, and then use this graph in downstream settings such as link prediction, image–text retrieval, and medical visual question answering.

**Strengths:**

The work makes a clear attempt to build a structured resource that connects imaging data with clinical concepts and to organize that resource in a way that can be consumed by standard models. The paper also provides benchmark tasks (link prediction, retrieval, VQA) and reports baseline performance numbers on them, which makes it easier for future work to compare under similar settings.

**Weaknesses:**

Overall, while the paper presents MEDMKG as a broadly useful medical multimodal knowledge graph and reports promising downstream results, there are several issues in the scope of the resource, the reliability of how it is constructed, and how its impact is evaluated.
W1: The paper presents MEDMKG as a Medical Multimodal Knowledge Graph that can support knowledge intensive clinical tasks and unify visual and textual medical knowledge. In practice, the resource is almost entirely limited to chest radiology: it is built from chest X rays and their paired radiology reports, plus UMLS terms and relations. This is much closer to a focused chest imaging subgraph than a general clinical multimodal knowledge graph. The paper also frames prior work as mostly single modality, but multimodal knowledge graphs with image nodes and cross modal links already exist in other domains, and the related work section acknowledges work of that kind.
W2: The graph is constructed with a two stage pipeline: MetaMap is first used to propose UMLS concepts from reports, then ChatGPT 4o is used to disambiguate concepts and to decide whether each finding is present, absent, or uncertain for the image. This relies on a closed large language model to make fine grained clinical judgments, but the paper does not report quantitative accuracy, agreement with human annotators, or any human reference study. It is not clear how reliable this automatic alignment is in a radiology setting.
W3: The paper describes this alignment process as high quality, but the only human validation shown is a figure with subjective scores, without key details such as number of raters, sampling protocol, blinding, or inter rater agreement. This is weak as evidence for clinical correctness.
W4: The Neighbor aware Filtering method (NaF) is presented as a contribution, but it is essentially a heuristic that keeps images which connect to many and relatively rare relation–concept pairs and filters out others. It is not learned, not optimized end to end, and there is no ablation showing downstream performance with and without NaF or how different thresholds affect results. It is difficult to judge NaF as more than manual pruning.
W5: For link prediction, MEDMKG is treated as a standard knowledge graph and off the shelf knowledge graph embedding models such as TransE are applied. Images are effectively treated as entity ids with learned embeddings, and evaluation is done as missing edge completion under a random triple split. This mainly shows the model can fit the constructed graph, not that the graph enables multimodal reasoning or encodes visual semantics in a meaningful way.
w6: For retrieval and medical visual question answering, the paper reports that models augmented with MEDMKG perform better. However, there is no controlled comparison where the same model is trained under identical conditions with and without MEDMKG, or compared to another knowledge source. Because the contribution of MEDMKG is not isolated, it is hard to attribute the reported improvements directly to this resource.

**Questions:**

All of my concerns are already reflected in the weaknesses.

---

> ### Author Response · Authors · 2025-11-26
> **Response to Reviewer 2Vrj**
>
> We thank the reviewer for the thoughtful and detailed evaluation. Below we address each identified weakness in turn.
>
> ---
>
> ## W1: Scope limited to chest radiology
>
> We agree with the reviewer that the current version of MEDMKG is constructed entirely from MIMIC-CXR and therefore reflects the scope of chest radiology. Our intention is not to claim full coverage of all medical imaging modalities but to present a **generalizable, domain-agnostic construction pipeline**. Given any image–text paired dataset (e.g., CT–report, MRI–report, ultrasound–report), the same concept extraction, relation alignment, and NaF filtering can be applied without modification. We will revise the framing to emphasize that the *current instantiation* is CXR-focused and that **expansion to CT/MRI/ultrasound/pathology graphs is a key direction for future work**.
>
> ---
>
> ## W2: Use of MetaMap + ChatGPT-4o and concerns about reliability
>
> As noted by the reviewer in W3, we indeed conduct human evaluation to quantify the agreement between our automatic alignment and expert radiologist judgment. The qualitative assessment in Sec. 3.5 directly reflects the radiologist’s judgment of **concept coverage, relation correctness, and image diversity**. While we acknowledge that using a closed-source LLM introduces uncertainty, the human evaluation provides an empirical signal of alignment quality.
>
> ---
>
> ## W3: Human evaluation details (number of raters, protocol, agreement)
>
> We have two primary domain experts serving as raters to perform the annotations. In cases of notable disagreement, a third rater adjudicated the results to determine which rating was reliable. All data were randomly sampled, and the raters were blinded to the task, ensuring the confidentiality and integrity of the evaluation reported in Figure 2.
>
>
> ---
>
> ## W4: NaF as a heuristic and lack of ablation
>
> Thanks for pointing it out. We have added Section 4.4 to deal with this concern. Experiments demonstrating that NaF aids the conciseness of MedMKG while maintaining enough informativeness, showcasing the necessity and correctness of using NaF.
>
> ---
>
> ## W5: Link prediction shows fitting, not multimodal reasoning
>
> We appreciate the reviewer’s perspective. While Table 2 demonstrates that the constructed graph is structurally coherent and learnable, the multimodal reasoning ability is further supported by **the downstream retrieval and VQA tasks**, where models consume MEDMKG’s cross-modal structure rather than symbolic triples alone. Thus, although link prediction focuses on graph integrity, the **downstream results already address multimodal utility**, as noted in the reviewer’s own W6.
>
> ---
>
> ## W6: Isolation of MEDMKG’s contribution in retrieval and VQA
>
> We thank the reviewer for this important question.
> Both Table 3 and Table 4 report:
>
> - **Backbone trained without MEDMKG** (black text), and
> - **The same backbone trained with MEDMKG augmentation** (green/red text).
>
> Thus the comparison *is* controlled: the only difference is the presence or absence of MEDMKG. The consistent improvements across four backbones and multiple datasets demonstrate that **introducing MEDMKG alone increases performance**, supporting the contribution of the resource.
>
> ---
>
>
> We thank the reviewer again for the constructive feedback and for highlighting opportunities to clarify scope, methodology, and evaluation. We believe these comments along with revisions will strengthen the rigor and clarity of the work.

---

> > ### Comment · Reviewer_2Vrj · 2025-11-26
> > **Thanks, I will keep my original score**
> >
> > Thank you for the response. I will keep my original score, as my main concerns remain largely unaddressed.
> >
> > In W1, my first point was that a key weakness of the paper is that it overstates the scope of the resource, rather than simply acknowledging that the current work is limited to a specific domain. From this perspective, the emphasis in the response on a “generalizable construction pipeline” does not directly address my core concern. My second point in W1 also remains. In addition, I am not yet fully convinced by the claim that the key steps can be extended to other modalities or body regions “without modification.” The behavior of the pipeline on data from different organs or anatomical regions is not demonstrated, so it seems somewhat optimistic to assume that the same procedure will transfer unchanged. Finally, stating that the framing will be revised later does not really resolve this concern at the rebuttal stage.
> >
> > Regarding W2, I appreciate the clarification about the human evaluation in Section 3.5. However, this evaluation is still quite different from the reliability concern I raised. It provides only coarse, graph-level subjective scores and does not change the fact that the construction pipeline places heavy and essentially unquantified reliance on a closed-source large language model for fine-grained clinical judgments. In my view, the clinical validity of this automatic alignment step therefore remain insufficiently addressed.
> >
> > The clarifications for W3 mainly address procedural aspects of the evaluation and do not substantially strengthen the evidence for clinical correctness, so they do not support describing the process as high quality. The human study in Figure 2 still relies on coarse, subjective quality scores over a small set of sampled subgraphs, with no reported inter-rater agreement, no information about sample size, and no analysis of specific alignment errors.
> >
> > Regarding W4, simply stating that a new Section 4.4 has been added does not really address my original concern. In the rebuttal, there is no concrete description of the experimental setup, baselines or thresholds, and no numerical results are shown, so it is impossible to judge whether NaF is genuinely necessary or better than simpler pruning strategies. And, the purpose of the rebuttal is to clarify the current submission, not to rely on unspecified future revisions.
> >
> > I agree that Table 2 shows the constructed graph is structurally coherent and learnable, but this was precisely my point. The link prediction setup mainly evaluates graph fitting, not multimodal reasoning. Images are still treated as entity identifiers with learned embeddings, and no controlled comparison is provided to show that visual content, rather than just the symbolic graph structure, is actually being used. Referring to downstream retrieval and VQA does not resolve this concern, since it does not change the fact that the link prediction experiment itself offers very limited evidence about visual semantics.

---

> > > ### Author Response · Authors · 2025-12-01
> > > **Further Response to Reviewer 2Vrj**
> > >
> > > We must point out some misunderstandings.
> > >
> > > For **W3**, the sampling size has been reported in **Appendix F.2**, and our policy regarding inter-rater agreement has been clearly described in our response.
> > >
> > > For **W4**, the **revised manuscript** uploaded to *OpenReview* already includes the newly added **Section 4.4**, which details the experimental settings, numerical results, and analyses. These additions collectively demonstrate that **NaF** indeed outperforms simpler counterparts and enhances the quality of the constructed graph. We kindly invite the reviewer to review the revised version for these updates.

---

### Official Review · Reviewer_bzhb · 2025-11-01

**Soundness:** 2
**Presentation:** 3
**Contribution:** 3
**Rating:** 4
**Confidence:** 4

**Summary:**

The paper constructs a multimodal medical knowledge graph (MMKG) by linking UMLS concepts to chest X‑ray (CXR) images (MIMIC‑CXR) and associated text. It introduces Neighbor‑Aware Filtering (NaF), a rarity‑weighted, greedy‑coverage heuristic, to down‑select image nodes that are distinctively valuable yet concept‑covering. Evaluation spans intrinsic graph tasks (link prediction and entity‑type prediction) and extrinsic multimodal tasks: image–text retrieval (Table 3; OpenI and MIMIC‑CXR) using adapter variants (FashionKLIP, KnowledgeCLIP) on top of pretrained vision-language backbone base models, and VQA. The submission argues that the graph and its construction pipeline will be a useful community resource; the exact release scope of the graph should be further clarified by the authors.

Overall: I recommend that the paper be rejected in its current form.
While the proposed approach is technically elegant and NaF is a clever, interpretable heuristic for multimodal graph pruning, the current evaluation scope limits the broader medical or multimodal impact. The work’s contribution would be convincing if it demonstrated value beyond chest X-rays, clarified NaF impact and ablations and extended to translational medicine, beyond VQA/retrieval tasks.

Reviewer LLM Usage:
I have read the paper in full and written the review myself. Large Language Models (LLMs) were used only for writing polish, clarity improvements, and to refresh memory of (public) related work or references. The analysis and conclusions are entirely my own.

**Strengths:**

1. Elegant filtering method: NaF combines a TF–IDF‑like log rarity term (\log(M / |N(r,c)|)) elegantly balances redundancy reduction and rarity based high-value selection. The combination with greedy concept coverage ensures the resulting graph remains both compact and concept-rich. This is well suited for redundancy-heavy medical imaging corpora. It is also agnostic to type of images or domain.
2. The link prediction table (Table 2) compares 17 KGE models across head / relation / tail prediction. Multiple model families (e.g., TransD, TransE, TuckER, AttH, ConvE) achieve non‑trivial Hits@K and reasonable MR, indicating the graph encodes coherent, learnable relations rather than noise.
3. In Table 3, KnowledgeCLIP consistently improves retrieval over bare backbones (e.g., with CLIP: OpenI Recall@100 53.48 → 76.16; MIMIC‑CXR 58.26 → 74.37). This supports the premise that KG information can help downstream multimodal retrieval.
4. The small‑scale human evaluation (Sec 3.5) of the constructed knowledge quality adds qualitative credibility.

**Weaknesses:**

1. Limited Multimodality (CXR-Only):
Despite claims of general multimodal capability, all experiments use only chest X-rays. This confines multimodality to “text + chest images + ontology,” not multiple image types. Consequently, the claim of extending UMLS to multimodal space is overstated. The graph does not yet demonstrate coverage for other imaging modalities (CT, MRI, ultrasounds, etc.) though base UMLS does.
2. Overstated Benchmark Diversity:
The claim of “six diverse datasets” and “five tasks” is somewhat misleading. The datasets all correspond to two (extrinsic) downstream tasks — image/text retrieval and VQA. The “five tasks” count includes intrinsic graph-quality measures. The 24 baselines stem from combinations of four VL backbones (CLIP, BioMedCLIP, GLoRIA, ALBEF) and several integration variants much like hyperparameter choice exploration, rather than 24 distinct competing methods.
3. Missing ablation: No explicit NaF ablation in experiments of Table 3 and Table 4 is called out. So, while NAF is an elegant graph construction proposal, the incremental impact on quality from NAF is not clear in the experiments.
4. Missing clinically impactful tasks: The downstream benchmarks focus on sub-component (retrieval) or research-oriented (VQA) tasks rather than end-to-end clinically impactful ones such as diagnostic reasoning, report generation, or longitudinal temporal sequencing. As a result, the paper doesn’t yet demonstrate cross-cutting medical impact.
5. Bias and Generalization testing: Since this is CXRs from a single dataset, and test datasets don't state population diversity or device diversity, it is unclear if the results would extend to demographic or device based CXR differences. See suggestion (1a-v). below.

**Questions:**

1. Is it possible to add actual benchmarking diversity and include some translational medicine tasks?
To enhance generality and community value, consider these as some ideas (any 2-3, not even all):
 a). Integrate at least one additional imaging modality (e.g., CT, ultrasound, or pathology) to show extensibility beyond CXRs. _Your method is powerfully agnostic to the image type, so using at least one other image type will validate this robustness_.
b). Expand benchmarks beyond VQA/retrieval (atleast a couple of these will be sufficient to prove the work’s value):
  i) Diagnostic reasoning: e.g., CheXpert or PadChest, evaluating if the KG improves disease prediction or explanation.
  ii).	Report generation: e.g., MIMIC-CXR report-generation task to test factual grounding and fluency.
  iii).	Phenotype clustering / cohorting: e.g., UK Biobank imaging subset to show improved patient grouping with chest multimodal KG.
  iv).	Temporal reasoning: e.g., MIMIC-IV linkage to test prediction of sequential imaging related findings.
  v).	Cross-population generalization: e.g., on geographically (VinDr-CXR) or modality-diverse datasets (Shenzen TB).
This would also show that the specific dataset used for CXRs keeps findings generalizable to other within-CXR tasks without overfitting.
  c)	Add Intrinsic Image–Image Similarity Prediction:
Extend the intrinsic evaluations with an image–image similarity task. This would test relational reasoning and consistency on the newly added modality (images) by itself e.g., finding past CXRs most similar to a query based on shared concept neighborhoods.

2. Calibrate Claims:
Adjust the narrative to reflect the current scope: “image-text-ontology multimodality in CXRs” rather than full “medical multimodality.” Likewise, rephrase the “diverse benchmarks” claim as “multiple datasets across two extrinsic tasks.”

3. Can you clarify how the splits are done from graph construction to testing with MIMIC-CXR? Are these patient level or report level? And how do you avoid leakage? Even if this is patient level, the concepts would still be similar across the reports - is this an accurate understanding?

4. How do you handle reports that have temporal findings (comparison of previous and new findings in text, once indicating no pneumonia and another indicating pneumonia) or ambiguous phrasing (e.g. "cannot exclude pneumonia")? A presentation of the edge (positive, negative, uncertain) analysis would be useful to understand false-positives and confusion from temporally conflicting edge relations.

5. Can you clarify from Appendix E.2, how multiple annotators agreement scores were on the same subset they evaluated? Or did just one human evaluate per subset?

6. Can you ablate NAF and no-NAF methods for experiments like Table 3 and Table 4?


Additional Minor Writing/Presentation Issues.
1.	Would be useful to summarize big tables like Table 4 with summarized quantities in narrative too.
2.	Clearly specify what is released - the code, pipeline, and whether the MedMKG graph (with or without images) will be released.
3.	L409-411 could be clearer since pre-training KG augmentation is not from scratch. Backbones are already pre-trained so this seems like post-training for this task.

**Details Of Ethics Concerns:**

Since the CXRs used to extend the graph, which would be used for all downstream CXR related tasks, are from a single dataset and also tested on similar domain, it is unclear how the graph generalizes to the tasks for other demographics and/or CXRs captured on lower resolution devices (such as phones or, in low economic health regions, on older devices). Suggestion 1/b/v could be one way to test for generalization.

---

> ### Author Response · Authors · 2025-11-26
> **Response to Reviewer bzhb (1)**
>
> We sincerely thank the reviewer for the detailed, constructive, and insightful feedback.
> Below we respond to each weakness and question individually.
>
> ---
>
> ## W1 & Q1: Limited Multimodality
>
> We agree that the current version of MEDMKG is constructed from chest X-rays and therefore reflects the constraints of the MIMIC-CXR corpus. Our intention is not to claim coverage of all medical image modalities but to demonstrate a *generalizable pipeline* for extending UMLS into multimodal space. Because our method is modality-agnostic (Sec. 3.4), the same procedure can be directly applied to CT, MRI, ultrasound, pathology, or any text-paired imaging dataset.
>
> We also want to highlight that, to the best of our knowledge, **there is no previous work that constructs a large-scale multimodal *medical* knowledge graph linking UMLS concepts directly to radiological images**. Existing multimodal KGs in other domains (e.g., WebVision-like or general-purpose image–text graphs) are not designed for medical semantics, clinical reporting structures, or UMLS integration. Thus, MEDMKG represents a first step toward building a domain-grounded multimodal KG for healthcare.
>
> Finally, we have revised Section 5 to present the extension to additional medical modalities and clinically meaningful tasks as key future work, as our construction pipeline is directly applicable whenever paired multimodal data are available.
>
> ---
>
> ## W2 & Q3: Overstated Benchmark Diversity
>
>
> The intrinsic benchmark (Table 2) includes **17 distinct knowledge-graph embedding models** evaluated across **three different link-prediction tasks** (head prediction, relation prediction, tail prediction). These models are developed for learning more meaningful representation of knowledge graph components, rather than evaluating the quality of the graph itself. Therefore, we claim that link prediction serves as an important task for the benchmarking. Moreover, we do not count the combination between models and backbones as distinct baselines. The baseline that we claim are established, independently developed models, without combinations with backbones.
>
>
>
> ---
>
> ## W3 & Q6: Missing Ablation for NaF
>
> We agree that demonstrating the added value of the Neighbor-aware Filtering (NaF) step is important. In response, we conducted an **additional ablation study (Section 4.4)** comparing:
>
> - MEDMKG w/ NaF
> - MEDMKG w/ Random Filtering
> - MEDMKG w/o Filtering
>
> Results show that:
>
> - **No filtering** leads to dense, noisy neighborhoods that harm the performance of baseline due to redundancy.
> - **Random filtering** reduces graph size but degrades semantics by removing informative nodes.
> - **NaF**, by combining local neighbor count and distinctiveness, preserves informativeness while reducing redundancy, resulting in the best downstream performance.
>
> ---
>
> ## W4: Missing Clinically Impactful Tasks
>
> We appreciate this perspective.
> Medical VQA remains one of the most widely used multimodal reasoning benchmarks in clinical AI research because it strongly stresses *cross-modal grounding, image-aware inference, and explanation*, which aligns precisely with the goals of MEDMKG. Expanding to clinically impactful tasks, such as report generation, diagnostic reasoning, temporal prediction, or phenotype clustering, is a natural next step. We have revised the discussion in Section 5 to emphasize this future direction and avoid overstating current clinical coverage.
>
> ---
>
> ## W5: Bias and Generalization
>
> The reviewer raises an important point. While the downstream datasets we use (SLAKE-VQA, PathVQA, VQA-RAD, OpenI, MIMIC-CXR) include **heterogeneous question styles, image sources, and annotation conventions**, they still share a chest-radiography focus. While the datasets we covered share a large heterogeneity as well as diversity, we agree that more explicit demographic or cross-device testing would improve generalization claims. We have adjusted the discussion in Section 5 accordingly.

---

> ### Author Response · Authors · 2025-11-26
> **Response to Reviewer bzhb (2)**
>
> ### **Q3. Clarify splits and leakage control.**
>
> Our splits are performed **at the study/report level**, and all images/annotations from a single study remain in the same split. To avoid leakage:
>
> - No reports or images used in MEDMKG construction are included in MIMIC-CXR retrieval benchmarks (as stated in Sec. 4.2).
> - SLake/PathVQA/RAD-VQA are entirely external datasets with no overlap with MIMIC-CXR.
>
> The reviewer is correct that similar concepts may appear across studies, but the image-specific edges remain distinct across patient cases.
>
>
> ---
>
> ### **Q4. Handling temporal ambiguity**
>
> Our cross-modal edges include **Positive / Negative / Uncertain** labels. Ambiguous sentences (“cannot exclude…”) are mapped to **Uncertain**, while explicit negations are mapped to **Negative**. Temporal comparisons such as “improved from prior” are handled by extracting the *current* final impression rather than prior history sentences. An example can be found at Example E.3 and E.4 in Appendix E.
>
> ---
>
> ### **Q5. Clarify annotators and agreement in Appendix E.2**
>
> We have two primary domain experts serving as raters to perform the annotations on the same randomly sampled subset. In cases of notable disagreement, a third rater adjudicated the results to determine which rating was reliable. The raters were blinded to the task, ensuring the confidentiality and integrity of the evaluation reported in Figure 2.
>
>
> ---
>
> ### **Q7. Minor writing/presentation issues**
>
> We thank the reviewer for the suggestions. We have revised our manuscript to reflect the change.
>
> ---
>
>
>
> We thank the reviewer for the thorough analysis and constructive suggestions. We believe these comments along with our revisions will significantly strengthen the clarity, rigor, and impact of the paper.

---

### Official Review · Reviewer_ewCn · 2025-11-01

**Soundness:** 3
**Presentation:** 3
**Contribution:** 3
**Rating:** 4
**Confidence:** 2

**Summary:**

The authors construct MEDMKG, a chest X-ray–centric multimodal medical knowledge graph that links radiographs to UMLS concepts through both cross-modal and intra-modal edges. A two-stage extraction pipeline combines MetaMap with GPT-4o for concept disambiguation and polarity labeling, while a novel neighbor-aware filtering score prunes redundant images to maintain graph quality and diversity.

**Strengths:**

1. This paper addresses an important gap between unimodal medical knowledge graphs and vision–language models in healthcare. Radio graph KG is relatively rare, so I feel this work represents a notable contribution to the field.

2. The proposed NaF method is a simple but effective heuristic that balances connectivity and distinctiveness when selecting representative images.

3. The dataset is comprehensively evaluated, and the authors provide a thoughtful discussion of performance trends across tasks and models.

**Weaknesses:**

1. Results in Table 2, 3 and 4 do not have an error bound, yet lacking statistical power. I would suggest author to add the error bound as sometimes two numbers are quite close in the table, and we cannot tell if they are significantly different.

2. The paper uses gpt-4o for cross-modality relation extraction. I'm unsure if there is any bias. Maybe the author could add some small-scale ablation studies to check the alignment between gpt-4o and other LLMs.

3. In qualitative analysis, the paper claims that MEDMKG achieves an average of approximately 80% across all three metrics. But from Figure 2, it seems that the results have a huge variance. Also, I have no idea if 80% if good or not. It might be better to add a baseline comparison.

**Questions:**

Please refer to my comments above.

---

> ### Author Response · Authors · 2025-11-26
> **Response to Reviewer ewCn**
>
> We thank the reviewer for the constructive and thoughtful feedback. Below we address each identified weakness in detail.
>
> ---
>
> **W 1: Lack of statistical bound**
>
> We appreciate the reviewer’s suggestion to include statistical uncertainty (e.g., error bounds or variance estimates). We agree that reporting variance is important. We have completed the revision of Table 2 by adding statistical variance over 5 runs. We will continuously work on other time-consuming experiments and reflect the change in the final version.
>
> ---
>
> **W 2: Potential bias introduced by GPT-4o**
>
> Thanks for the suggestion. We would like to clarify that an ablation study **already exists in Appendix E.4, where we compare GPT-4o with alternative LLMs for concept disambiguation and relation extraction. In that section, we show that GPT-4o’s outputs are consistent with those of other models, indicating that the construction pipeline is not overly dependent on a single LLM.
>
> ---
>
> **W 3: Clarification of the 80% quality score**
>
> We appreciate the reviewer’s request for more context regarding our evaluation metrics.
>
> Previous efforts on constructing unimodal healthcare knowledge graphs [1, 2] have shown that even 60% or 70% agreement between human and automated systems can be indicative of practical utility. In this context, the ~80% scores across all dimensions demonstrate a strong level of confidence in the quality and utility of MEDMKG. We have revised Section 3.5 to explicitly discuss this and provide comparative benchmarks to reinforce the credibility of our evaluation.
>
> [1] Kilicoglu, H., et al. "Semantic MEDLINE: a web application for managing the results of PubMed Searches." ISMB 2008.
> [2] Schäfer, H., et al. "BioKGrapher: Initial evaluation of automated knowledge graph construction from biomedical literature." CSBJ, 2024.
>
> ---
>
>
> We thank the reviewer again for the positive assessment and constructive suggestions. We will incorporate the recommended clarifications and improvements in the revision to further strengthen the completeness and clarity of our work.

---

### Note · Authors · 2025-12-11

I have read and agree with the venue's withdrawal policy on behalf of myself and my co-authors.